# Find Your Friends: Personalized Federated Learning with the Right Collaborators

## Abstract

In the traditional federated learning setting, a central server coordinates a network of clients to train one global model. However, the global model may serve many clients poorly due to data heterogeneity. This problem can be mitigated when participating clients learn personalized models that can better serve their own needs. By noting that each client's distribution can be represented as a mixture of all clients' distributions, we derive a principled algorithm based on expectation maximization. Our framework, FedeRiCo, estimates the utilities of other participants' models on each client's data so that everyone can select the right collaborators for learning. As a result, each client can learn as much or as little from other clients as is optimal for its local data distribution. Additionally, we theoretically analyze the convergence of FedeRiCo and empirically demonstrate its communication efficiency even in the fully decentralized setting. Our algorithm outperforms other federated, personalized, and/or decentralized approaches on several benchmark datasets, being the *only* approach that consistently performs better than training with local data alone.

## 1 Introduction

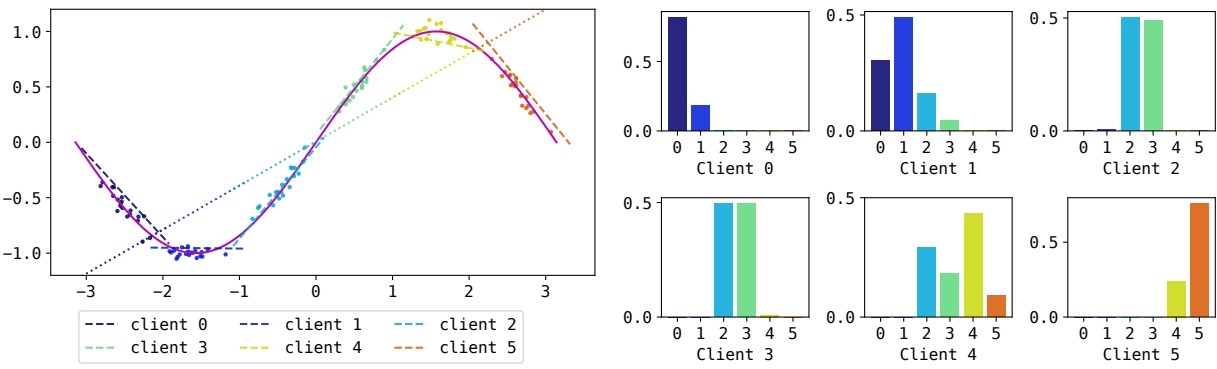

Figure 1: **Left:** Noisy data points generated for clients along a sine curve (solid) where the $x$ and $y$ axes are the input and target, respectively. The FedAvg model (dotted line) fails to adapt to the local data seen by each client, in contrast to our FedeRiCo personalized models (dashed lines). **Right:** The collaboration weights learned by FedeRiCo to average participant outputs for each client. Clients learn to collaborate only with similar neighbours.

Federated learning (FL) (McMahan et al., 2017) offers a framework in which a central server-side model is collaboratively trained across decentralized client datasets. This approach has been successfully implemented in practice for developing machine learning models without direct access to client data, which is crucial in heavily regulated industries such as banking and healthcare (Long et al., 2020; Sadilek et al., 2021). For example, multiple hospitals that collect patient data may desire to merge their datasets for increased diversity and size, but are unable due to privacy regulations.

Statistical heterogeneity (Zhao et al., 2018; Adnan et al., 2022) is a major and common practical challenge in FL, where each client may hold different data distributions. Traditional FL methods like Federated Averaging (FedAvg) (McMahan et al., 2017) have demonstrated promising performance with homogeneous client data. However, these methods often struggle to handle statistical heterogeneity for two main reasons. Firstly, the variation in client distributions can lead to divergences in weight updates during training (Zhao et al., 2018). Secondly, it can be challenging for a single global model to provide optimal performance across all clients during inference.

As an illustrative example, consider a simple scenario where each client seeks to fit a linear model to limited data on an interval of the sine curve as shown in Figure 1. This is analogous to the FL setting where several participating clients would like to collaborate, but each client only has access to data from its own data distribution. It is clear that no single linear model can be adequate to describe the entire joint dataset, so a global model learned by FedAvg can perform poorly, as shown by the dotted line. Ideally, each client should benefit from collaboration by increasing the effective size and diversity of data available, but in practice, forcing everyone to use the same global model without proper personalization can hurt performance on their own data distribution (Kulkarni et al., 2020; Tan et al., 2022).

To address this, we propose **Fede**rating with the **Ri**ght **Co**llaborators (FedeRiCo), a novel personalized framework suitable for every client to find other participants with similar data distributions to collaborate with. As illustrated in Figure 1, FedeRiCo enables each client to choose the *right collaborators*; clients are able to correctly leverage information from neighboring clients when it is beneficial to do so. The final personalized models can serve the local distributions well, as demonstrated in the top plot.

More specifically, FedeRiCo assumes that each client has an underlying data distribution, and exploits the hidden relationships among the clients' data. By selecting the most relevant clients, each client can collaborate as much or, maybe more importantly, as little as they need, and learn a personalized mixture model to fit the local data. Additionally, FedeRiCo can achieve this in a fully decentralized manner that is not beholden to any central authority (Li et al., 2021a; Huang et al., 2021; Kalra et al., 2023).

**Our contributions** We propose FedeRiCo, a novel personalized FL framework based on expectation-maximization (EM). We also show that with some common assumptions, our algorithm is guaranteed to converge. Through extensive experiments on several benchmark datasets, we demonstrate that our approach finds good client collaboration and outperforms other methods in the non-i.i.d. setting.

**Paper outline** The rest of the paper is organized as follows. In Section 2 we discuss related approaches towards personalized federated learning. Section 3 describes our algorithm formulation, its relationship to expectation-maximization, and an efficient protocol for updating clients. We provide experimental results in Section 4, and conclude in Section 5.

## 2 Related Work for Personalized Federated Learning

**Meta-learning** Federated learning can be interpreted as a meta-learning problem, where the goal is to extract a global meta-model based on data from several clients. This meta-model can be learned using, for instance, the well-known Federated Averaging (FedAvg) algorithm (McMahan et al., 2017), and personalization can then be achieved by locally fine-tuning the meta-model (Jiang et al., 2019). Later studies explored methods to learn improved meta-models. Khodak et al. (2019) proposed ARUBA, a meta-learning algorithm based on online convex optimization, and demonstrates that it can improve upon FedAvg's performance. Per-FedAvg (Fallah et al., 2020) uses the Model Agnostic Meta-Learning (MAML) framework to build the initial meta-model. However, MAML requires computing or approximating the Hessian term and can therefore be computationally prohibitive. Acar et al. (2021) adopted gradient correction methods to explicitly de-bias the meta-model from the statistical heterogeneity of client data and achieved sample-efficient customization of the meta-model.

**Model regularization / interpolation** Several works improve personalization performance by regularizing the divergence between the global and local models (Hanzely & Richtárik, 2020; Li et al., 2021b; Huang et al., 2021; Zhang et al., 2022). Similarly, PFedMe (Dinh et al., 2020) formulates personalization as a proximal regularization problem using Moreau envelopes. FML (Shen et al., 2020) adopts knowledge distillation

to regularize the predictions between local and global models and handle model heterogeneity. In recent work, SFL (Chen et al., 2022a) also formulates the personalization as a bi-level optimization problem with an additional regularization term on the distance between local models and its neighbor models according to a connection graph. Specifically, SFL adopts GCN to represent the connection graph and learns the graph as part of the optimization to encourage useful client collaborations. Introduced by Mansour et al. (2020) as one of the three methods for achieving personalization in FL, model interpolation involves mixing a client's local model with a jointly trained global model to build personalized models for each client. Deng et al. (2020) further derive generalization bounds for mixtures of local and global models.

**Multi-task learning** Personalized FL naturally fits into the multi-task learning (MTL) framework. MOCHA (Smith et al., 2017) utilizes MTL to address both systematic and statistical heterogeneity but is restricted to simple convex models. VIRTUAL (Corinzia et al., 2019) is a federated MTL framework for non-convex models based on a hierarchical Bayesian network formed by the central server and the clients, and inference is performed using variational methods. SPO (Cui et al., 2022) applies Specific Pareto Optimization to identify the optimal collaborator sets and learn a hypernetwork for all clients. While also aiming to identify necessary collaborators, SPO adopts a centralized FL setting with clients jointly training the hypernetwork. In contrast, our work focuses on decentralized FL where clients aggregate updates from collaborators, and jointly make predictions.

In a similar spirit to our work, Marfoq et al. (2021) assume that the data distribution of each client is a mixture of several underlying distributions/components. Federated MTL is then formulated as a problem of modeling the underlying distributions using Federated Expectation-Maximization (FedEM). Clients jointly update a set of several global models, also known as component models, and each maintains a customized set of weights for prediction, corresponding to the mixing coefficients of the underlying distributions. One shortcoming of FedEM is that it uses an instance-level weight assignment during training but a client-level weight assignment at inference time. As a concrete example, consider a client consisting of a 20%/80% data mixture from distributions A and B. FedEM will learn two models, one for each distribution. Given a new data point at inference time, the client will always predict $0.2 \cdot \text{pred}_A + 0.8 \cdot \text{pred}_B$, *regardless of whether the data point came from distribution A or B*. This is caused by the mismatched behaviour between training and inference time. On the contrary, FedeRiCo naturally considers a client-level weight assignment for both training and inference in a decentralized setting.

**Other approaches** Clustering-based approaches are also popular for personalized FL (Sattler et al., 2020; Ghosh et al., 2020; Mansour et al., 2020). Such personalization lacks flexibility since each client can only collaborate with other clients within the same cluster. FedFomo (Zhang et al., 2021) interpolates the model updates of each client with those of other clients to improve local performance. FedPer (Arivazhagan et al., 2019) divides the neural network model into base and personalization layers. Base layers are trained jointly, whereas personalization layers are trained locally. Self-FL (Chen et al., 2022b) balances local training and global training objectives from uncertainty perspective.

## 3 Federated Learning with the Right Collaborators

### 3.1 Problem Formulation

We consider a federated learning (FL) scenario with $K$ clients. Let $[K] := \{1, 2, \ldots, K\}$ denote the set of positive integers up to $K$. Each client $i \in [K]$ has a local dataset $D_i = \{(\mathbf{x}_s^{(i)}, y_s^{(i)})\}_{s=1}^{n_i}$ where $n_i$ is the number of examples for client $i$, and the input $\mathbf{x}_s \in \mathcal{X}$ and output $y_s \in \mathcal{Y}$ are drawn from a joint distribution $\mathcal{D}_i$ over the space $\mathcal{X} \times \mathcal{Y}$.

The goal of personalized FL is to find a prediction model $h_i : \mathcal{X} \mapsto \mathcal{Y}$ that can perform well on the local distribution $\mathcal{D}_i$ for each client. One of the main challenges in personalized FL is that we do not know if two clients $i$ and $j$ share the same underlying data distribution. If their data distributions are vastly different, forcing them to collaborate is likely to result in worse performance compared to local training without collaboration. Our method, **Fede**rating with the **Ri**ght **Co**llaborators (FedeRiCo), is designed to address this problem so that each client can choose to collaborate or not, depending on their data distributions. For better exposition, Section 3.2 first demonstrates how our algorithm works in the centralized setting with a

central server. Then Section 3.3 presents several enhancements of FedeRiCo so that it will work even in the fully decentralized setting with minimal communication overhead.

## 3.2   FedeRiCo in Centralized Settings

Note that every client's local distribution $\mathcal{D}_i$ can always be represented as a mixture of $\{\mathcal{D}_j\}_{j=1}^K$ with some weights $\boldsymbol{\pi}_i = [\pi_{i1}, \ldots, \pi_{iK}] \in \Delta^K$, where $\Delta^K$ is the $(K-1)$-dimensional simplex[1]. Let $z_i$ be the latent assignment variable of client $i$, and $\Pi := [\boldsymbol{\pi}_1, \ldots, \boldsymbol{\pi}_K]^\top$ be the prior $\Pi_{ij} = \Pr(z_i = j)$. Suppose that the conditional probability $p_i(y|\mathbf{x})$ satisfies $-\log p_i(y|\mathbf{x}) = \ell(h_{\boldsymbol{\phi}_i^*}(\mathbf{x}), y) + c$ for some parameters $\boldsymbol{\phi}_i^* \in \mathbb{R}^d$, loss function $\ell : \mathcal{Y} \times \mathcal{Y} \mapsto \mathbb{R}^+$, and normalization constant $c$. By

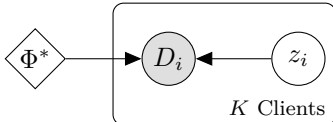

Figure 2: Graphical model

using the stacked notation $\Phi^* = [\boldsymbol{\phi}_1^*, \ldots, \boldsymbol{\phi}_K^*] \in \mathbb{R}^{d \times K}$, Figure 2 shows the graphical model of how the local dataset is generated. Our goal is to learn the parameters $\Theta := (\Phi, \Pi)$ by maximizing the log-likelihood:

$$f(\Theta) := \frac{1}{n} \log p(D; \Theta) = \frac{1}{n} \sum_{i=1}^K \log p(D_i; \Theta) = \frac{1}{n} \sum_{i=1}^K \log \sum_{z_i=1}^K p(D_i, z_i; \Theta). \tag{1}$$

where $D := \cup_i D_i$ and $n := \sum_i n_i$.

One standard approach to optimization with latent variables is expectation maximization (EM) (Dempster et al., 1977). The corresponding variational lower bound is given by[2]

$$\mathcal{L}(q, \Theta) := \frac{1}{n} \sum_i \mathbb{E}_{q(z_i)}[\log p(D_i, z_i; \Theta)] + C, \tag{2}$$

where $C$ is a constant not depending on $\Theta$ and $q$ is an alternative distribution. To obtain concrete objective functions suitable for optimization, we further assume that the marginal distributions satisfy $p_i(x) = p(x), \forall i \in [K]$. Similar to Marfoq et al. (2021), we adopted this assumption as it allows us to narrow our focus on discriminative modelling. With this assumption, we perform the following updates at each iteration $t$:

**E-step:** Calculate the client weights $w_{ij}$, which represent the latent variable assignment $z$ for each client by finding their best alternative distribution $q^*$:

$$w_{ij}^{(t)} := q^{*(t)}(z_i = j) \propto \Pi_{ij}^{(t-1)} \exp\left[-\sum_{s=1}^{n_i} \ell\left(h_{\boldsymbol{\phi}_j^{(t-1)}}(\mathbf{x}_s^{(i)}), \ y_s^{(i)}\right)\right]. \tag{3}$$

**M-step:** Given the posterior $q^{*(t)}$ from the E-step, maximize $\mathcal{L}$ w.r.t. $\Theta = (\Phi, \Pi)$:

$$\Pi_{ij}^{(t)} = w_{ij}^{(t)} \quad \text{and} \quad \Phi^{(t)} \in \operatorname*{argmin}_\Phi \sum_{i=1}^K \widehat{\mathcal{L}}_{w,i}(\Phi), \tag{4}$$

$$\text{where} \quad \widehat{\mathcal{L}}_{w,i}(\Phi) := \sum_{j=1}^K \frac{w_{ij}^{(t)}}{n_i} \sum_{s=1}^{n_i} \ell\left(h_{\boldsymbol{\phi}_j}(\mathbf{x}_s^{(i)}), \ y_s^{(i)}\right). \tag{5}$$

Note that the prior $\Pi_{ij}^{(t)}$ for the next iteration will be updated using the posterior from the current iteration $w_{ij}^{(t)}$. Due to the update in Equation (4), we refer to $\Pi_{ij}^{(t)}$ or $w_{ij}^{(t)}$ as client weights. For each client, the

---

[1]One-hot $\boldsymbol{\pi}_i$ is always feasible, but other mixtures may exist. When $\boldsymbol{\pi}_i$ is one-hot with a one in the $i$th position, each client learns alone without collaboration. While this "mixture" is trivial, it is an important case to have available for circumstances in which all other clients' distributions are truly different and collaboration would be detrimental.

[2]All derivations of this section can be found in Appendix A.

priors $\Pi_{ij}^{(t-1)}$ from previous round can be stored locally and used to update the posterior $w_{ij}^{(t)}$ once it receives the model $\Phi^{(t-1)}$ from the server. $\Phi^{(t)}$ in the M-step, however, is trickier to compute since each client can potentially update $\Phi$ towards different directions due to data heterogeneity amongst the clients. Bear in mind that each client can only see its local data $D_i$ in the federated setting. To stabilize optimization and avoid overfitting from client updates, we rely on small gradient steps in lieu of full optimization in each round. Since solving for $\Phi^{(t)}$ exactly is computationally expensive, we approximate it using gradient descent:

1. Each participating client computes the local gradient $\nabla\widehat{\mathcal{L}}_{w,i}(\Phi^{(t-1)})$ on its local dataset $D_i$ with fixed weights $w_{ij}^{(t)}$, and sends the gradient back to the server.
2. The server aggregates the gradients and updates the central model using a step size $\eta > 0$:

$$\Phi^{(t)} = \Phi^{(t-1)} - \eta \sum_{j=1}^{K} \frac{n_j}{n} \cdot \nabla\widehat{\mathcal{L}}_{w,j}(\Phi^{(t-1)}). \tag{6}$$

Finally, at inference time, each client uses $\widehat{h}_i(\mathbf{x}) = \sum_j w_{ij}^{(t)} h_{\phi_j^{(t)}}(\mathbf{x})$ for prediction.

**Remark 1** The posterior $w_{ij}^{(t)}$ (or equivalently the prior in the next iteration $\Pi_{ij}^{(t)}$) reflects the importance of model $\phi_j$ on the data $D_i$. When $w_{ij}^{(t)}$ is one-hot with a one in the $i$th position, client $i$ can perform learning by itself without collaborating with others. When $w_{ij}^{(t)}$ is more diverse, client $i$ can find the right collaborators with useful models $\phi_j$. Such flexibility enables each client to make its own decision on whether or not to collaborate with others, hence the name of our algorithm.

**Remark 2** Unlike prior work (Mansour et al., 2020; Marfoq et al., 2021), our assignment variable $z$ and probability $\Pi$ are on the client level. If we assume that all clients share the same prior (i.e., there is only a vector $\boldsymbol{\pi}$ instead of a matrix $\Pi$), the algorithm would be similar to HypCluster (Mansour et al., 2020). Marfoq et al. (2021) used a similar formulation as ours but their assignment variable $z$ is on the instance level: every data point (instead of every client) comes from a mixture of distributions. Such an approach can cause several issues at inference time, as the assignment for a novel data point is unknown. We refer the interested readers to Section 2 and Section 4 for further comparison.

**Theoretical convergence** Under some regularity assumptions, our algorithm converges as follows:

**Theorem 3.1.** *[Convergence] Under Assumptions B.1-B.6, when the clients use SGD with learning rate $\eta = \frac{a_0}{\sqrt{T}} > 0$, and the number of rounds $T \geq a_0^2 \cdot \max\{8L^2, 16L^2\beta^2\}$, the iterates of our algorithm satisfy*

$$\frac{1}{T}\sum_{t=1}^{T} \mathbb{E}\|\nabla_\Phi f(\Phi^t, \Pi^t)\|_F^2 \leq \mathcal{O}\left(\frac{1}{\sqrt{T}}\right), \tag{7}$$

$$and \quad \frac{1}{T}\sum_{t=1}^{T} \mathbb{E}[\Delta_\Pi f(\Phi^t, \Pi^t)] \leq \mathcal{O}\left(\frac{1}{T^{3/4}}\right), \tag{8}$$

*where the expectation is over the random batch samples and $\Delta_\Pi f(\Phi^t, \Pi^t) := f(\Phi^t, \Pi^t) - f(\Phi^t, \Pi^{t+1}) \geq 0$.*

Due to their length, the technical assumptions, further details, and the complete proof are deferred to Appendix B. The above theorem shows that the gradient with respect to the model parameters $\Phi$ and the improvement over the mixing coefficients $\Pi$ becomes small as we increase the number of rounds $T$, thus converging to a stationary point of the negative log-likelihood objective $f$.

### 3.3 A Communication-Efficient Protocol in Decentralized Settings

So far, we have discussed how FedeRiCo works in the centralized setting, however, this setting presents several challenges (Lian et al., 2017; Li et al., 2020). Especially for cross-silo FL, there may not exist a third party that all clients trust, and that has the computational resources to act as a central server (Kalra et al., 2023). Centralization creates a single point of failure for both the ongoing communication between clients, and for client privacy via data leaks (Beltrán et al., 2022). To surmount these challenges, we propose several enhancements for FedeRiCo to work in the fully decentralized setting with minimal communication.

Specifically, we tackle the bottlenecks in both the E-step (3) and the M-step (4) since they require joint information of all models $\Phi$. The pseudocode of the complete algorithm is provided in Algorithm 1, which is the implementation we use in our experiments.

**E-step** For client $i$, the key missing quantity to compute (3) are the losses $\ell(\phi_j^{(t-1)})$, or likelihoods $p(D_i|z_i = j; \Phi^{(t-1)})$, of other clients' models $\phi_j, j \neq i$. Since the models $\Phi$ are being updated slowly, one can expect that $\ell(\phi_j^{(t-1)})$ will not be significantly different from the loss $\ell(\phi_j^{(t-2)})$ of the previous iteration. Therefore, each client can maintain a list of losses for all the clients, sample a subset of clients in each round using a sampling scheme $\mathcal{S}$ (e.g., $\epsilon$-greedy sampling as discussed later), and only update the losses of the chosen clients. In our experiment, we show that even sampling only **one** other client would be sufficient, making FedeRiCo have the same communication requirement on average as some centralized algorithms like FedAvg.

**M-step** To make clear how $\Phi$ is updated in the M-step, we focus on the update to a specific client's model $\phi_i$. According to (5) and (6), the update to $\phi_i$ is given by

$$-\eta \sum_{j=1}^{K} w_{ji}^{(t)} \sum_{s=1}^{n_j} \nabla_{\phi_i} \ell \left( h_{\phi_i}(\mathbf{x}_s^{(j)}), \ y_s^{(j)} \right). \tag{9}$$

Note that the aggregation is based on $w_{ji}^{(t)}$ instead of $w_{ij}^{(t)}$. Intuitively, this suggests $\phi_i$ should be updated based on how the model is being used by *other clients* rather than how client $i$ itself uses it. If $\phi_i$ does not appear to be useful to any clients, i.e. $w_{ji}^{(t)} = 0, \ \forall j$, it does not get updated. Therefore, whenever client $i$ is sampled by another client $j$ using the sampling scheme $\mathcal{S}$, it will send $\phi_i$ to $j$, and receive the gradient update $\mathbf{g}_{ij} := w_{ji}^{(t)} \sum_{s=1}^{n_j} \nabla_{\phi_i} \ell(h_{\phi_i}(\mathbf{x}_s^{(j)}), \ y_s^{(j)})$ from client $j$. One issue here is that $\mathbf{g}_{ij}$ is governed by $w_{ji}^{(t)}$, which could be arbitrarily small, leading to no effective update to $\phi_i$. We will show how this can be addressed by using an $\epsilon$-greedy sampling scheme.

**Sampling scheme $\mathcal{S}$** We deploy an $\epsilon$-greedy scheme where, in each round, each client uniformly samples clients with probability $\epsilon \in [0, 1]$ and samples the client(s) with the highest posterior(s) otherwise. This allows a trade off between emphasizing gradient updates from high-performing clients (small $\epsilon$), versus receiving updates from clients uniformly to find potential collaborators (large $\epsilon$). The number $M$ of sampled clients (neighbors) per round and $\epsilon$ can be tuned based on the specific problem instance. We will show the effect of varying the hyperparameters in the experiments.

**Tracking the losses for the posterior** The final practical consideration is the computation of the posterior $w_{ij}^{(t)}$. From the E-step (3) and the M-step (4), one can see that $w_{ij}^{(t)}$ is the softmax transformation of the negative accumulative loss $L_{ij}^{(t)} := \sum_{\tau=1}^{t-1} \ell_{ij}^{(\tau)}$ over rounds (see Appendix A for derivation). However, the accumulative loss can be sensitive to noise and initialization. If one of the models, say $\phi_j$, performs slightly better than other models for client $i$ at the beginning of training, then client $i$ is likely to sample $\phi_j$ more frequently, thus enforcing the use of $\phi_j$ even when better models exist. To address this, we instead keep track of the exponential moving average of the loss with a momentum parameter $\beta \in [0, 1)$, $\widehat{L}_{ij}^{(t)} = (1 - \beta)\widehat{L}_{ij}^{(t-1)} + \beta l_{ij}^{(t)}$, and compute $w_{ij}^{(t)}$ using $\widehat{L}_{ij}^{(t)}$. This encourages clients to seek new collaborators rather than focusing on existing ones.

---

**Algorithm 1:** FedeRiCo: Federating with the Right Collaborators

---

**Input:** Client local datasets $\{D_i\}_{i=1}^K$, number of communication rounds $r$, number of neighbors $M$, $\epsilon$-greedy sampling probability $\epsilon$, momentum for exponential moving average loss tracking $\beta$, learning rate $\eta$.

**Output:** Client models $\{\phi_i\}_{i=1}^K$ and client weights $w_{ij}$.

    `// Initialization`

**1** Randomly initialize $\{\phi_i\}_{i=1}^K$;

**2** **for** *client $C_i$ in $\{C_i\}_{i=1}^K$* **do**

**3**     Initialize $\widehat{L}_{ij}^{(0)} = 0, \ell_{ij}^{(0)} = 0, w_{ij}^{(0)} = \frac{1}{K}$;

**4** **end**

**5** **for** *iterations $t = 1 \ldots T$* **do**

**6**     **for** *client $C_i$ in $\{C_i\}_{i=1}^K$* **do**

**7**        Sample $M$ neighbors of this round $B^t$ according to $\epsilon$-greedy selection w.r.t. $w_{ij}^{(t-1)}$;

**8**        Send $\phi_i$ to other clients that sampled $C_i$;

**9**        Receive $\phi_j$ from sampled neighbors $B^t$;

          `// E-step`

**10**        $\ell_{ij}^{(t)} = \ell_{ij}^{(t-1)}$ ;                         `// Keep the loss from previous round`

**11**        **for** *$b$ in $B^t$* **do**

**12**           $\ell_{ib}^{(t)} = \sum_{s=1}^{n_i} \ell\left(h_{\phi_b^{(t)}}(\mathbf{x}_s^{(i)}),\ y_s^{(i)}\right)$ ;                `// Update the sampled ones`

**13**        **end**

**14**        $\widehat{L}_{ij}^{(t)} = (1 - \beta)\widehat{L}_{ij}^{(t-1)} + \beta\ell_{ij}^{(t)}$ ;        `// Update exponential moving averages`

**15**        $w_{ij}^{(t)} = \frac{\exp(-\widehat{L}_{ij}^{(t)})}{\sum_{j'=1}^K \exp(-\widehat{L}_{ij'}^{(t)})}$;

          `// M-step`

**16**        **for** *$C_b$ in $B^t$* **do**

             `// Could also do multiple gradient steps instead`

**17**           Compute and send $\mathbf{g}_{bi} = w_{ib}^{(t)}\nabla_{\phi_b}\sum_{s=1}^{n_i}\ell\left(h_{\phi_b}(\mathbf{x}_s^{(i)}),\ y_s^{(i)}\right)$ to $C_b$;

**18**        **end**

**19**        **for** *$C_j$ that sampled $C_i$* **do**

**20**           Receive $\mathbf{g}_{ij} = w_{ji}^{(t)}\sum_{s=1}^{n_j}\nabla_{\phi_i}\ell\left(h_{\phi_i}(\mathbf{x}_s^{(j)}),\ y_s^{(j)}\right)$;

**21**        **end**

**22**        $\phi_i^t = \phi_i^{(t-1)} - \eta\sum_j \mathbf{g}_{ij}$ ;               `// Or any other gradient-based method`

**23**     **end**

**24** **end**

---

## 4 Experiments

### 4.1 Experimental Settings

We conduct a range of experiments to evaluate the performance of our proposed FedeRiCo with multiple datasets. Additional experiment details and results can be found in Appendix C.

**Datasets** We compare different methods on several real-world datasets. We evaluate on image classification tasks with the CIFAR-10, CIFAR-100 (Krizhevsky et al., 2009), and Office-Home[3] (Venkateswara et al., 2017) datasets. Particularly, we consider a non-IID data partition among clients by first splitting data by labels into several groups with disjoint label sets. Each group is considered a distribution, and each client samples from one distribution to form its local data. For each client, we randomly divide the local data into 80% training data and 20% test data.

**Baseline methods** We compare our FedeRiCo to several federated learning baselines. FedAvg (McMahan et al., 2017) trains a single global model for every client. We also compare to other personalized FL approaches including FedAvg with local tuning (FedAvg+) (Jiang et al., 2019), Clustered FL (Sattler et al., 2020), FedEM (Marfoq et al., 2021)[4], FedFomo (Zhang et al., 2021), as well as a local training baseline. All accuracy results are reported as the mean and standard deviation across different random data splits and random training seeds. Unless specified otherwise, we allow each client to communicate with 3 other clients (neighbours) per round, with $\epsilon = 0.3$ and momentum $\beta = 0.6$ as the default hyperparameters for FedeRiCo in all experiments. For FedEM, we let all clients train 4 components jointly for the learned distribution, which provides sufficient capacity to accommodate different numbers of label groups (or data distributions). For FedFomo, we hold out 20% of the training data for client weight calculations. For FedAvg+, we follow Marfoq et al. (2021) and update the local model with 1 epoch of local training.

**Training settings** For all models, we use the Adam optimizer with learning rate 0.01. CIFAR experiments use 150 rounds of training, while Office-Home experiments use 400 rounds. CIFAR-10 results are reported across 5 different data splits and 3 different training seeds for each data split. CIFAR-100 and Office-Home results are reported across 3 different data splits each with a different training seed.

### 4.2 Performance Comparison

Table 1: Accuracy (in percentage) with different number of data distributions. Best results in bold.

| Method | CIFAR-10 # of distributions | | | CIFAR-100 # of distributions | | | Office-Home # of distributions | | |
|---|---|---|---|---|---|---|---|---|---|
| | 2 | 3 | 4 | 2 | 3 | 4 | 2 | 3 | 4 |
| FedAvg | $11.44_{\pm 3.28}$ | $11.73_{\pm 3.68}$ | $13.93_{\pm 5.74}$ | $21.28_{\pm 5.04}$ | $17.41_{\pm 3.27}$ | $18.36_{\pm 3.68}$ | $66.58_{\pm 1.88}$ | $53.36_{\pm 4.21}$ | $51.25_{\pm 4.37}$ |
| FedAvg+ | $12.45_{\pm 8.46}$ | $29.86_{\pm 17.85}$ | $45.65_{\pm 21.61}$ | $29.95_{\pm 1.07}$ | $35.33_{\pm 1.77}$ | $36.17_{\pm 3.27}$ | $80.21_{\pm 0.68}$ | $81.88_{\pm 0.91}$ | $84.50_{\pm 1.37}$ |
| Local Training | $40.09_{\pm 2.84}$ | $55.27_{\pm 3.11}$ | $69.03_{\pm 7.05}$ | $16.60_{\pm 0.64}$ | $25.99_{\pm 2.38}$ | $31.05_{\pm 1.68}$ | $76.76_{\pm 0.23}$ | $83.30_{\pm 0.32}$ | $88.05_{\pm 0.44}$ |
| Clustered FL | $11.50_{\pm 3.65}$ | $15.24_{\pm 5.79}$ | $16.43_{\pm 5.17}$ | $20.93_{\pm 3.57}$ | $23.15_{\pm 7.04}$ | $15.15_{\pm 0.60}$ | $66.58_{\pm 1.88}$ | $53.36_{\pm 4.21}$ | $51.25_{\pm 4.37}$ |
| FedEM | $41.21_{\pm 10.83}$ | $55.08_{\pm 6.71}$ | $63.61_{\pm 9.93}$ | $26.25_{\pm 2.40}$ | $24.11_{\pm 7.36}$ | $19.23_{\pm 2.58}$ | $22.59_{\pm 1.95}$ | $28.72_{\pm 1.83}$ | $22.46_{\pm 3.99}$ |
| FedFomo | $42.24_{\pm 8.32}$ | $59.45_{\pm 5.57}$ | $71.05_{\pm 6.09}$ | $12.15_{\pm 0.57}$ | $20.49_{\pm 2.90}$ | $24.53_{\pm 2.77}$ | $78.61_{\pm 0.78}$ | $82.57_{\pm 0.24}$ | $87.86_{\pm 0.77}$ |
| FedeRiCo | $\mathbf{56.61}_{\pm 2.51}$ | $\mathbf{69.76}_{\pm 2.25}$ | $\mathbf{78.22}_{\pm 4.80}$ | $\mathbf{30.95}_{\pm 1.62}$ | $\mathbf{39.19}_{\pm 1.64}$ | $\mathbf{41.41}_{\pm 1.07}$ | $\mathbf{83.56}_{\pm 0.49}$ | $\mathbf{90.28}_{\pm 0.75}$ | $\mathbf{93.76}_{\pm 0.12}$ |

The performance of each FL method is shown in Table 1. Following the settings introduced by Marfoq et al. (2021), each client is evaluated on its own local testing data and the average accuracies weighted by local dataset sizes are reported. We observe that FedeRiCo has the best performance across all datasets and number of data distributions. Here, local training can be seen as an indicator to assess if other methods benefit from client collaboration as local training has no collaboration at all. We observe that our proposed FedeRiCo is the only method that consistently outperforms local training, meaning that FedeRiCo is the only method that consistently encourages effective client collaborations. Notably, both FedEM and FedFomo performs comparably well to FedeRiCo on CIFAR-10 but worse when the dataset becomes more complex like

---

[3]This dataset is publically available for research purposes only.
[4]We use implementations from `https://github.com/omarfoq/FedEM` for Clustered FL and FedEM.

CIFAR-100. This indicates that building the right collaborations among clients becomes a harder problem for more complex datasets. Moreover, FedEM can become worse as the number of distributions increases, even worse than local training, showing that it is increasingly hard for clients to participate effectively under the FedEM framework for complex problems with more data distributions.

In addition, Clustered FL has similar performance to FedAvg, indicating that it is hard for Clustered FL to split into the right clusters. In Clustered FL (Sattler et al., 2020), every client starts in the same cluster and cluster only split when the FL objective is close to a stationary point, i.e. the norm of averaged gradient update from all clients inside the cluster is small. Therefore, in a non-i.i.d setting like ours, the averaged gradient update might always be noisy and large, as clients with different distributions are pushing diverse updates to the clustered model. As a result, cluster splitting rarely happens which makes clustered FL behave similarly to FedAvg in practice.

### 4.3 Client Collaboration

In this section, we investigate client collaboration by plotting the personalized client weights $w_{ij}^{(t)}$ of FedeRiCo over training. With different client data distributions, we show that FedeRiCo assigns more weight to similar clients coming from the same distribution. As observed in Figure 3, similar clients collaborate to make the final predictions. For example, clients 3, 4 and 7 use a mixture of predictions from each other (in light blue) while client 0 only uses itself for prediction as it is the only client coming from distribution 0 (in dark blue) in this particular random split.

On the contrary, as shown in Figure 4, even with 4 components, FedEM fails to use all of them for predictions for the 4 different data distributions. Notably, clients 2, 3, 4, 6 and 7 coming from two different distributions are using only the component 3 for prediction, while component 0 is never used by any client. Based on this, we find FedeRiCo better encourages the clients to collaborate with other similar clients and less with different clients. Each client can collaborate as much or as little as they need. Additionally, as all the non-similar clients have a weight of (almost) 0, each client only needs a few models for prediction.

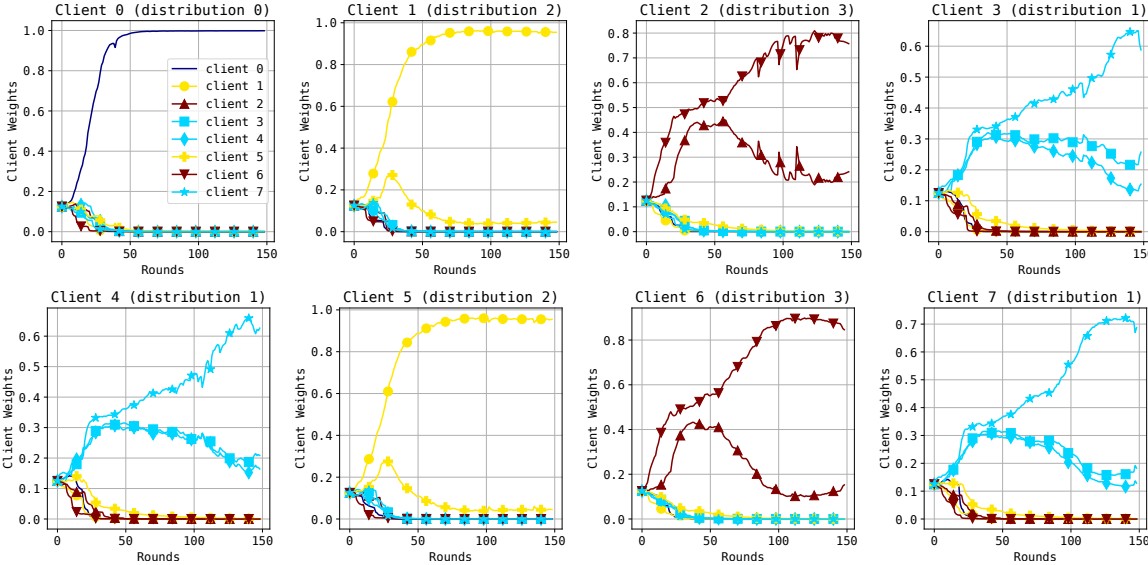

Figure 3: Client weights over time of FedeRiCo with CIFAR100 data and four different client distributions. Clients are color coded by their private data's distribution.

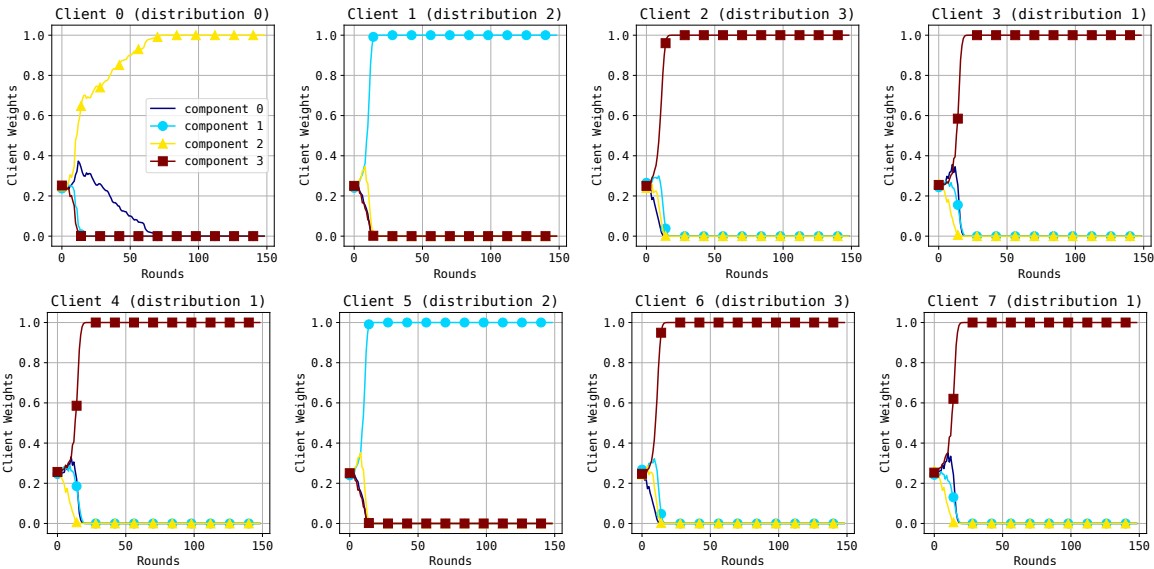

Figure 4: Component weights over training for FedEM with 4 components, on CIFAR100 data with 4 different client distributions. Clients are color coded by their private data's distribution.

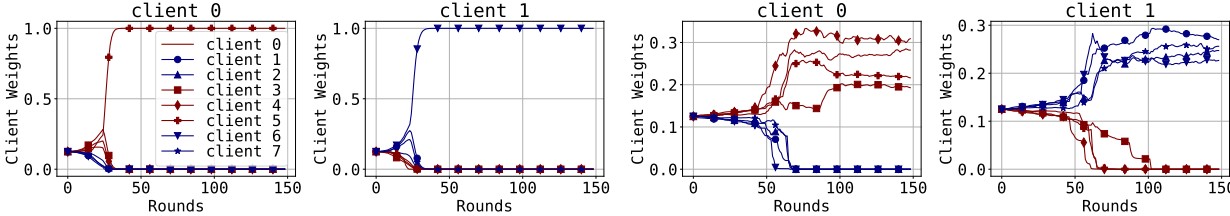

Figure 5: Client weights on CIFAR-10 with two different client distributions. **Left two:** Client weights with accumulative loss. **Right two:** Client weights with exponential moving average.

### 4.4 Effect of Using Exponential Moving Average Loss

Here, we visualize the effect of using the exponential moving average loss by plotting client weights with both accumulative loss and exponential moving average loss in Figure 5.[5] We observe that with the accumulative loss, the client weights quickly converge to one-hot, while with the exponential moving average loss, the client weights are more distributed to similar clients. This corresponds to our expectation stated in Section 3.3: the clients using exponential moving average loss are expected to seek for more collaboration compared to using accumulative loss. It is also worth noting that although the client weight vectors become one-hot part way through training with the accumulative loss, clients are not learning in isolation. In fact, clients 0, 3, 4, and 5, who share a common data distribution, all end up using client 5's model only, whereas clients 1, 2, 6, and 7 end up using client 6's model only. We show the weights of all clients in Figure 8 of Appendix D.

### 4.5 Hyperparameter Sensitivity

In this section, we explore the effect of hyperparameters of our proposed FedeRiCo.

**Effect of $\epsilon$-greedy sampling** Here we show the effect of different $\epsilon$ values. Recall that each client deploys an $\epsilon$-greedy selection strategy. The smaller the value of $\epsilon$, the more greedy the client is in selecting the most relevant collaborators with high weights, leading to less exploration. Figure 6 shows the accuracy over training with different $\epsilon$ values on the Office-Home dataset. One can see that there is a trade-off between exploration and exploitation. If $\epsilon$ is too high (e.g., $\epsilon = 1$, uniform sampling), then estimates of the

---

[5]We used uniform sampling for the accumulative loss ($\epsilon = 1$) as most of the client weights are 0 after a few rounds.

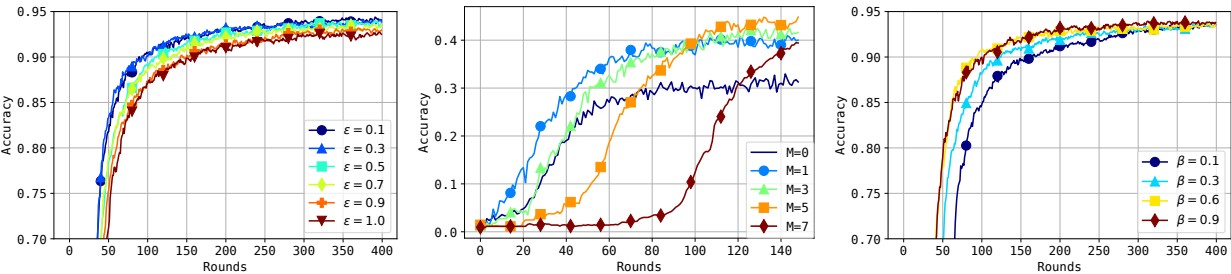

Figure 6: Test accuracy with different hyperparameters: sampling $\epsilon$ (**left**), number of sampled neighbors (**middle**), momentum $\beta$ (**right**).

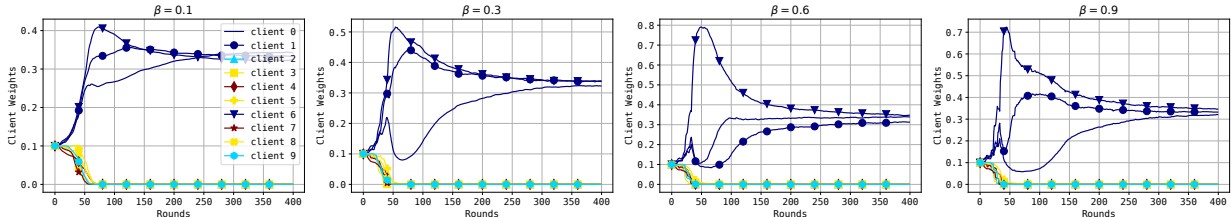

Figure 7: Client weights learned by client 0 with different momentum values $\beta$ on the client weight update.

likelihoods/losses are more accurate. However, some gradient updates will vanish because the client weight is close to zero (see Section 3.3), resulting in slow convergence. On the other hand, if $\epsilon$ is too small, the client may miss some important collaborators due to a lack of exploration. As a result, we use a moderate $\epsilon = 0.3$ in all experiments.

**Effect of number of sampled neighbors**  We plot accuracy with number of neighbors $M \in \{0, 1, 3, 5, 7\}$ on CIFAR100 using 4 different client distributions, where $M = 0$ is similar to Local Training as no collaboration happens. As shown in Figure 6, when the number of neighbors increases, FedeRiCo converges more slowly as each client is receiving more updates on other client's models. While a smaller number of neighbors seems to have a lower final accuracy, we notice that even with $M = 1$, we still observe significant improvement compared to no collaboration. Therefore, we use $M = 3$ neighbors in our experiments as it has reasonable performance and communication cost.

**Effect of client weight momentum**  We plot the overall test accuracy of client 0 on the Office-Home dataset with 4 different data distributions over $\beta \in \{0.1, 0.3, 0.6, 0.9\}$ in Figure 6 and similarly for the client weights in Figure 7. With smaller $\beta$, as shown in Figure 7, we observe a smoother update on the client weights, which is expected as the old tracking loss dominates the new one. Although various values produce similar final client weights, a bigger $\beta$ can lead to more drastic changes in early training. However, one shouldn't pick a very small $\beta$ just because it can produce smoother weights. As shown in Figure 6, the algorithm may converge more slowly with smaller $\beta$. Therefore, we use $\beta = 0.6$ as it encourages smoother updates and also maintains good convergence speed.

In Appendix D, Table 4 shows the effect of using a Dirichlet data split, which is a common choice in FL (Marfoq et al., 2021). Our method still outperforms all benchmarks. Finally, in Figure 10, we consider whether there is a benefit to better optimization in the M-step of our method by increasing the number of local epochs. While there is some improvement when using more optimization, there is a tradeoff with increased computation time.

## 5   Conclusion and Discussion

In this paper, we proposed FedeRiCo, a novel framework for personalized FL derived from EM for non-i.i.d data. We evaluated FedeRiCo across different datasets and demonstrated that FedeRiCo outperforms

multiple personalized FL baselines and encourages clients to collaborate with similar clients, i.e., the right collaborators.

**Federated learning architectures** Traditional FL methods, such as Federated Averaging (FedAvg) (McMahan et al., 2017), primarily focus on a centralized FL architecture, where a single central server coordinates and aggregates updates from all clients. However, this centralized setting is heavily dependent on the central server and is thus subject to several challenges in practice. Firstly, the presence of a commonly trusted central server may not always be guaranteed. Additionally, the server may fail or be maliciously attacked, disrupting the entire system (Li et al., 2020). Furthermore, as previously mentioned, when all clients simultaneously communicate with the server, the communication burden on the server can be substantial (Lian et al., 2017). To mitigate these challenges, decentralized FL (Dai et al., 2022; Beltrán et al., 2022) has emerged as a viable solution for addressing single point failure and reducing the communication bandwidth on the server.

**Security implications** Security is a significant concern in FL frameworks, particularly regarding malicious attacks (Blanco-Justicia et al., 2021), which may be further exacerbated in a decentralized setting without the central authority. One potential solution is to integrate existing trust mechanisms, such as blockchain (Qin et al., 2022; Short et al., 2020) and homomorphic encryption (Nguyen & Thai, 2022), with FL frameworks (Kairouz et al., 2021) to ensure secure collaboration.

**Privacy implications** A key motivation of federated learning's original development was to protect clients' privacy by avoiding sharing private data (McMahan et al., 2017). However, FL schemes, centralized or decentralized, do not provide an explicit guarantee of privacy since sharing model parameters (or updates) may also reveal important information about the data (Carlini et al., 2019; Lyu et al., 2022). To address these concerns, a promising direction is to apply differentially private mechanisms in FL training to provide privacy guarantees (Wei et al., 2020; Truex et al., 2020; Liu et al., 2022), which invoke a trade-off between utility and privacy (Chen et al., 2022c; Bietti et al., 2022). Fortunately, most of the existing FL frameworks, including FedeRiCo are compatible with differential privacy.

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

## A  Derivations

**Variational lower bound**  Here we derive the variational lower bound Equation (2) for the log-likelihood objective Equation (1). For each $i \in [K]$,

$$\log \sum_{z_i=1}^{K} p(D_i, z_i; \Theta) = \log \sum_{z_i=1}^{K} q(z_i) \cdot \frac{p(D_i, z_i; \Theta)}{q(z_i)} \tag{10}$$

$$= \log \mathbb{E}_{q(z_i)} \left[ \frac{p(D_i, z_i; \Theta)}{q(z_i)} \right] \tag{11}$$

$$\geq \mathbb{E}_{q(z_i)} \left[ \log \frac{p(D_i, z_i; \Theta)}{q(z_i)} \right] \tag{12}$$

$$= \mathbb{E}_{q(z_i)} \left[ \log p(D_i, z_i; \Theta) \right] - \mathbb{E}_{q(z_i)} [\log q(z_i)], \tag{13}$$

where $q$ is an alternative distribution, the inequality is due to Jensen's Inequality and the last term $\mathbb{E}_{q(z_i)}[\log q(z_i)]$ is constant independent of the parameter $\Theta$.

**Derivations of the EM steps**  Given the assumptions in the main text about $p_i(y|\mathbf{x})$ and $p_i(\mathbf{x})$, we know that

$$- \log p(D_i | z_i = j; \Phi) = \sum_{s=1}^{n_i} \ell(h_{\phi_j}(\mathbf{x}_s)^{(i)}, y_s^{(i)}) - \log p(\mathbf{x}_s^{(i)}) + c. \tag{14}$$

- **E-step:** Find the best $q^*$ for each client given the current parameters $\Theta^{(t-1)}$:

$$w_{ij}^{(t)} := q^{*(t)}(z_i = j) = p(z_i = j | D_i; \Theta^{(t-1)}) \tag{15}$$

$$= \frac{p(z_i = j | \Pi^{(t-1)}) \cdot p(D_i | z_i = j; \Phi^{(t-1)})}{\sum_{j'=1}^{K} p(z_i = j' | \Pi^{(t-1)}) \cdot p(D_i | z_i = j'; \Phi^{(t-1)})} \tag{16}$$

$$= \frac{\Pi_{ij}^{(t-1)} \cdot p(D_i | z_i = j; \Phi^{(t-1)})}{\sum_{j'=1}^{K} \Pi_{ij'}^{(t-1)} \cdot p(D_i | z_i = j'; \Phi^{(t-1)})} \tag{17}$$

$$\propto \Pi_{ij}^{(t-1)} \exp \left[ -\sum_{s=1}^{n_i} \ell\left( h_{\phi_j^{(t-1)}}(\mathbf{x}_s^{(i)}), \ y_s^{(i)} \right) \right]. \tag{18}$$

Then the variational lower bound becomes

$$\mathcal{L}(q^{(t)}, \Theta) = \frac{1}{n} \sum_i \sum_j w_{ij}^{(t)} \cdot \log p(D_i, z_i = j; \Theta) + C \tag{19}$$

$$= \frac{1}{n} \sum_i \sum_j w_{ij}^{(t)} \cdot \left( \log p(z_i = j; \Pi) + \log p(D_i | z_i = j; \Phi) \right) + C \tag{20}$$

$$= \frac{1}{n} \sum_i \sum_j w_{ij}^{(t)} \cdot \left( \log \Pi_{ij} + \log p(D_i | z_i = j; \Phi) \right) + C. \tag{21}$$

- **M-step:** Given the posterior $w_{ij}^{(t)}$ from the E-step, we need to maximize $\mathcal{L}$ w.r.t. $\Theta = (\Phi, \Pi)$. For the priors $\Pi$, we can optimize each row $i$ of $\Pi$ individually since they are decoupled in Equation (21). Note that each row of $\Pi$ is also a probability distribution, so the optimum solution is given by $\Pi_{ij}^{(t)} = w_{ij}^{(t)}$. This is because the first term of Equation (21) for each $i$ is the negative cross entropy, which is maximized when $\Pi_{ij}$ matches $w_{ij}^{(t)}$.
  Optimizing Equation (21) w.r.t. $\Phi$ gives

$$\Phi^{(t)} \in \underset{\Phi}{\operatorname{argmax}} \mathcal{L}(q^{(t)}, \Theta) = \underset{\Phi}{\operatorname{argmin}} \frac{1}{n} \sum_{i=1}^{K} \sum_{j=1}^{K} w_{ij}^{(t)} \sum_{s=1}^{n_i} \ell\left( h_{\phi_j}(\mathbf{x}_s^{(i)}), \ y_s^{(i)} \right). \tag{22}$$

**Posterior and accumulative loss** Here we show an alternative implementation for Equation (3) using accumulative loss. To shorten notations, let $\ell_{ij}^{(t)} := \sum_{s=1}^{n_i} \ell\left(h_{\phi_j^{(t)}}(\mathbf{x}_s^{(i)}), y_s^{(i)}\right)$. Combining Equation (3) and Equation (4) gives

$$w_{ij}^{(t)} = p(z_i = j | D_i; \Theta^{(t-1)}) \tag{23}$$

$$\propto w_{ij}^{(t-1)} \exp\left[-\ell_{ij}^{(t-1)}\right] \tag{24}$$

$$\propto w_{ij}^{(t-2)} \exp\left[-\left(\ell_{ij}^{(t-2)} + \ell_{ij}^{(t-1)}\right)\right]. \tag{25}$$

We can see that it is accumulating the losses of previous models (e.g., $\phi_j^{(t-2)}$, $\phi_j^{(t-1)}$ and so on) inside the exponential. Therefore, assuming the uniform prior $\Pi_{ij}^{(0)} = 1/K, \forall j$, $w^{(t)}$ is the softmax transformation of the negative of the accumulative loss $L_{ij}^{(t)} := \sum_{\tau=1}^{t-1} \ell_{ij}^{(\tau)}$ up until round $t$.

## B  Convergence Proof

This section provides the missing proof of our convergence result (Theorem 3.1). At a high level, we will show that our algorithm in Section 3.2 can be viewed as an instantiation of the Federated Surrogate Optimization (Marfoq et al., 2021, Algo.3). Therefore, we can apply the general convergence result of Federated Surrogate Optimization (Marfoq et al., 2021, Thm.3.2′) to get our convergence guarantee. Toward that end, Lemma B.7 shows that the local objective

$$f_i(\Theta) = f_i(\Phi, \pi_i) := -\frac{1}{n_i} \log p(D_i | \Phi, \pi_i) = -\frac{1}{n_i} \sum_{s=1}^{n_i} \log p(x_i^{(s)}, y_i^{(s)} | \Phi, \pi_i), \tag{26}$$

admits a *partial first-order surrogates* near $(\Phi^{(t-1)}, \Pi^{(t-1)})$ (Marfoq et al., 2021, Def.1), given by

$$g_i^{(t)}(\Phi, \Pi) := g_i^{(t)}(\Phi, \boldsymbol{\pi}_i) \tag{27}$$

$$:= \frac{1}{n_i} \sum_{s=1}^{n_i} \sum_{j=1}^{K} q_j^{(t)} \left[\ell\left(h_{\phi_j}(x_i^{(s)}), y_i^{(s)}\right) - \log p_j(x_i^{(s)}) - \log \pi_{ij} + \log q_j^{(t)} - c\right] \tag{28}$$

so that our algorithm shares the same form as Federated Surrogate Optimization. Then Theorem 3.1 follows from Marfoq et al. (2021, Thm.3.2′) under suitable assumptions.

To start, we adapt assumptions 2 to 7 of Marfoq et al. (2021) to our setting as follows:

**Assumption B.1.** $\forall i \in [K], p_i(x) = p(x)$.

**Assumption B.2.** The conditional probability $p_i(y|x)$ satisfies

$$-\log p_i(y|x) = \ell(h_{\phi_i^*}(x), y) + c, \tag{29}$$

for some parameters $\phi_i^* \in \mathbb{R}^d$, loss function $\ell : \mathcal{Y} \times \mathcal{Y} \mapsto \mathbb{R}^+$ and normalization constant $c$.

Let $f(\Phi, \Pi) := \frac{1}{n} \log p(D; \Phi, \Pi)$ be the log-likelihood objective as in Equation (1).

**Assumption B.3.** $f$ is bounded below by $f^* \in \mathbb{R}$.

**Assumption B.4** (Smoothness and bounded gradient)**.** For all $x, y$, the function $\phi \mapsto \ell(h_\phi(x), y)$ is $L$-smooth, twice continuously differentiable and has bounded gradient: there exists $B < \infty$ such that $\|\nabla_\phi \ell(h_\phi(x), y)\| \le B$.

**Assumption B.5** (Unbiased gradients and bounded variance)**.** Each client $i \in [K]$ can sample a random batch $\xi$ and compute an unbiased estimator $\mathbf{g}_i(\phi, \xi)$ of the local gradient with bounded variance, i.e., $\mathbb{E}_\xi[\mathbf{g}_i(\phi, \xi)] = \frac{1}{n_i} \sum_{s=1}^{n_i} \nabla \ell(h_\phi(\mathbf{x}_i^{(s)}), y_i^{(s)})$ and $\mathbb{E}_\xi \|\mathbf{g}_t(\phi, \xi) - \frac{1}{n_i} \sum_{s=1}^{n_i} \nabla \ell(h_\phi(\mathbf{x}_i^{(s)}), y_i^{(s)})\| \le \sigma^2$.

**Assumption B.6** (Bounded dissimilarity). There exist $\beta$ and $G$ such that any set of weights $\boldsymbol{\gamma} \in \Delta^K$:

$$\sum_{i=1}^{K} \frac{n_i}{n} \left\| \frac{1}{n_i} \sum_{s=1}^{n_i} \sum_{j=1}^{K} \gamma_j \nabla \ell(h_{\boldsymbol{\phi}}(\mathbf{x}_i^{(s)}), y_i^{(s)}) \right\|^2 \leq G^2 + \beta^2 \left\| \frac{1}{n} \sum_{i=1}^{K} \sum_{s=1}^{n_i} \sum_{j=1}^{K} \gamma_j \nabla \ell(h_{\boldsymbol{\phi}}(\mathbf{x}_i^{(s)}), y_i^{(s)}) \right\|^2. \tag{30}$$

**Lemma B.7** (Partial first-order surrogate). $g_i^{(t)}$ *is a partial first-order surrogate of* $f_i$ *near* $(\Phi^{(t-1)}, \Pi^{(t-1)})$.

*Proof.* In the following, we will verify that $g_i^{(t)}$ satisfies the three conditions of partial first-order surrogates near $(\Phi^{(t-1)}, \Pi^{(t-1)})$:

1. $g_i^{(t)}(\Phi, \Pi) \geq f_i(\Phi, \Pi), \forall t, \Phi, \Pi$;

2. $r_i^{(t)}(\Phi, \Pi) := g_i^{(t)}(\Phi, \Pi) - f_i(\Phi, \Pi)$ is differentiable and $\widetilde{L}$-smooth w.r.t. $\Phi$ (for some $\widetilde{L} < \infty$). Moreover, $r_i^{(t)}(\Phi^{(t-1)}, \Pi^{(t-1)}) = 0$ and $\nabla_{\Phi} r_i(\Phi^{(t-1)}, \Pi^{(t-1)}) = \mathbf{0}$;

3. $g_i^{(t)}(\Phi, \Pi^{(t-1)}) - g_i(\Phi, \Pi) = \mathsf{d}(\Pi^{(t-1)}, \Pi)$ for all $\Phi$ and $\Pi \in \arg\min_{\Pi'} g(\Phi, \Pi')$ where $\mathsf{d}$ is non-negative and $\mathsf{d}(\Pi, \Pi') = 0$ iff $\Pi = \Pi'$.

To simplify notations, define the following (the dependency on round $t$ is ignored when it is clear from context)

$$q_j := q_i(z_i = j), \tag{31}$$

$$\mathcal{L}_j := \sum_{s=1}^{n_i} \ell\left(h_{\boldsymbol{\phi}_j}(x_i^{(s)}), y_i^{(s)}\right), \tag{32}$$

$$\gamma_j := p_i(z_i = j | D_i, \Phi, \boldsymbol{\pi}_i). \tag{33}$$

**Condition 1** To start verifying the first condition,

$$g_i(\Phi, \boldsymbol{\pi}_i) = \frac{1}{n_i} \sum_{s=1}^{n_i} \sum_{j=1}^{K} q_j \left[ \ell\left(h_{\boldsymbol{\phi}_j}(x_i^{(s)}), y_i^{(s)}\right) - \log p_j(x_i^{(s)}) - \log \pi_{ij} + \log q_j - c \right] \tag{34}$$

$$= \frac{1}{n_i} \sum_{s=1}^{n_i} \sum_{j} q_j \left[ -\log\left(p_j(y_i^{(s)}|x_i^{(s)}, \boldsymbol{\phi}_j) \cdot p_j(x_i^{(s)}) \cdot p_i(z_i = j)\right) + \log q_j \right] \tag{35}$$

$$= \frac{1}{n_i} \sum_{s=1}^{n_i} \sum_{j} q_j \left[ -\log p_i\left(x_i^{(s)}, y_i^{(s)}, z_i = j \Big| \Phi, \boldsymbol{\pi}_i\right) + \log q_j \right] \tag{36}$$

$$= \frac{1}{n_i} \sum_{j} q_j \left[ -\log p_i\left(D_i, z_i = j | \Phi, \boldsymbol{\pi}_i\right) + \log q_j \right]. \tag{37}$$

Then

$$r_i(\Phi, \boldsymbol{\pi}_i) = g_i(\Phi, \boldsymbol{\pi}_i) - f_i(\Phi, \boldsymbol{\pi}_i) \tag{38}$$

$$= \frac{1}{n_i} \mathcal{KL}\left( q(\cdot) \parallel p_t(\cdot|D_i, \Phi, \boldsymbol{\pi}_i) \right), \tag{39}$$

where $\mathcal{KL}$ is the KL-divergence. This verifies the first condition of partial first-order surrogates since the KL-divergence is non-negative.

**Condition 2** Now we verify the second condition. Note that $r_t$ is twice continuously differentiable due to Assumption B.4. With Assumption B.1

$$\gamma_j = p_i(z_i = j | D_i, \Phi, \boldsymbol{\pi}_i) = \frac{\exp\left[-\mathcal{L}_{j'} + \log \pi_{ij}\right]}{\sum_{j'} \exp\left[-\mathcal{L}_{j'} + \log \pi_{ij'}\right]}, \tag{40}$$

$$\nabla_{\boldsymbol{\phi}_{j'}} \gamma_j = \begin{cases} (-\gamma_j + \gamma_j^2)\nabla\mathcal{L}_j & \text{if } j' = j \\ \gamma_j \gamma_{j'} \nabla\mathcal{L}_{j'} & \text{if } j' \neq j, \end{cases} \tag{41}$$

where $\nabla\mathcal{L}_j$ is shorthand for $\nabla_{\boldsymbol{\phi}_j}\mathcal{L}_j$. Then

$$\nabla_{\boldsymbol{\phi}_{j'}} r_i = \frac{1}{n_i}\nabla_{\boldsymbol{\phi}_{j'}}\sum_j(-q_j \log \gamma_j) \qquad\qquad \text{Definition of } \mathcal{KL} \tag{42}$$

$$= \frac{1}{n_i}\sum_j\left(-\frac{q_j}{\gamma_j}\nabla_{\boldsymbol{\phi}_{j'}}\gamma_j\right) \tag{43}$$

$$= \frac{1}{n_i}\left[q_{j'}(1 - \gamma_{j'}) - \sum_{j \neq j'} q_j \gamma_{j'}\right]\nabla\mathcal{L}_{j'} \qquad \text{When } j = j' \text{ vs } j \neq j' \tag{44}$$

$$= \frac{1}{n_i}\left[q_{j'}(1 - \gamma_{j'}) - (1 - q_{j'})\gamma_{j'}\right]\nabla\mathcal{L}_{j'} \qquad \sum_j q_j = 1 \tag{45}$$

$$= \frac{1}{n_i}(q_{j'} - \gamma_{j'})\nabla\mathcal{L}_{j'}. \tag{46}$$

The Hessian of $r_i$, $\mathbf{H}(r_i) \in \mathbb{R}^{dK \times dK}$ w.r.t. $\Phi$, is a block matrix, with blocks given by

$$\left(\mathbf{H}(r_t)\right)_{j,j'} = \begin{cases} \frac{1}{n_i}\left[(q_j - \gamma_j)\mathbf{H}(\mathcal{L}_j) + (\gamma_j - \gamma_j^2)(\nabla\mathcal{L}_j)(\nabla\mathcal{L}_j)^\top\right] \\ -\frac{1}{n_i}\gamma_j \gamma_{j'}(\nabla\mathcal{L}_j)(\nabla\mathcal{L}_{j'})^\top & \text{when } j \neq j', \end{cases} \tag{47}$$

where $\mathbf{H}(\mathcal{L}_j) \in \mathbb{R}^{d \times d}$ is the Hessian of $\mathcal{L}_{\boldsymbol{\phi}_j}(D_t)$ w.r.t. $\boldsymbol{\phi}_j$. Introduce block matrices $\widetilde{\mathbf{H}}, \widehat{\mathbf{H}} \in \mathbb{R}^{dK \times dK}$ as

$$\widetilde{\mathbf{H}}_{j,j'} = \begin{cases} \frac{1}{n_i}(\gamma_j - \gamma_j^2)(\nabla\mathcal{L}_j)(\nabla\mathcal{L}_j)^\top \\ -\frac{1}{n_i}\gamma_j\gamma_{j'}(\nabla\mathcal{L}_j)(\nabla\mathcal{L}_{j'})^\top & \text{when } j \neq j', \end{cases}$$
$$\widehat{\mathbf{H}}_{j,j'} = \begin{cases} \frac{1}{n_i}(q_j - \gamma_j)\mathbf{H}(\mathcal{L}_j) \\ \mathbf{0} & \text{when } j \neq j'. \end{cases} \tag{48}$$

Since $q_j, \gamma_j \in [0, 1]$ and $\ell$ is $L$-smooth by Assumption B.4, we have $-L \cdot I_{dK} \preccurlyeq \widehat{\mathbf{H}} \preccurlyeq L \cdot I_{dK}$. Using Lemma B.8 (see below), we have $\mathbf{0} \preccurlyeq \widetilde{\mathbf{H}} \preccurlyeq B^2 \cdot I_{dK}$ (note that $\nabla\mathcal{L}_j$ is the sum of $n_i$ individual gradients and $\mathbf{H}(r_t)$ has $1/n_i$). As a result, $-\widetilde{L} \cdot I_{dK} \preccurlyeq \mathbf{H}(r_t) \preccurlyeq \widetilde{L} \cdot I_{dK}$ (where $\widetilde{L} = L + B^2 < \infty$) and therefore $r_t$ is $\widetilde{L}$-smooth.

Finally, $q_j^{(t)} = p_i(z_i = j | D_i, \Phi^{(t-1)}, \boldsymbol{\pi}_i^{(t-1)}), \forall t > 0$ by the algorithm, which means

$$r_i^{(t)}(\Phi^{(t-1)}, \Pi^{(t-1)}) = r_i^{(t)}(\Phi^{(t-1)}, \boldsymbol{\pi}_i^{(t-1)}) = 0. \tag{49}$$

Additionally, from Equation (39) we know that $r_i^{(t)}(\Phi, \boldsymbol{\pi}_i)$ is a (non-negative) KL-divergence for all $\Phi, \Pi$. Recall that $r_i^{(t)}$ is differentiable. It follows that $\Phi^{(t-1)}$ is a minimizer of the function $\{\Phi \mapsto r_i^{(t)}(\Phi, \boldsymbol{\pi}_i^{(t-1)})\}$ and

$$\nabla_\Phi r_i^{(t)}(\Phi^{(t-1)}, \boldsymbol{\pi}_i^{(t-1)}) = \mathbf{0}. \tag{50}$$

This verifies the second condition of the partial first-order surrogate.

**Condition 3** Note that $\boldsymbol{\pi}_i^{(t)} = \arg\min_{\boldsymbol{\pi}} g_i^{(t)}(\Phi, \boldsymbol{\pi})$ due to the choice of $q_i^{(t)}$ by the algorithm. Then for any $\boldsymbol{\pi}_i$ and $i \in [K]$,

$$
\begin{aligned}
g_i^{(t)}(\Phi, \boldsymbol{\pi}_i) - g_i^{(t)}(\Phi, \boldsymbol{\pi}_i^{(t)}) &= \sum_j q_j^{(t)}(\log \pi_{ij}^{(t)} - \log \pi_{ij}) \\
&= \sum_j \pi_{ij}^{(t)}(\log \pi_{ij}^{(t)} - \log \pi_{ij}) \\
&= \mathcal{KL}(\boldsymbol{\pi}_i^{(t)} \| \boldsymbol{\pi}_i),
\end{aligned}
\tag{51}
$$

which is non-negative and equals zero iff $\boldsymbol{\pi}_i^{(t)} = \boldsymbol{\pi}_i$. This verifies the third condition of partial first-order surrogate. $\qquad\square$

The following lemma is used when verifying the second condition from above.

**Lemma B.8.** *Suppose* $\mathbf{g}_1, \ldots, \mathbf{g}_K \in \mathbb{R}^d$ *and* $\boldsymbol{\gamma} = (\gamma_1, \ldots, \gamma_K) \in \Delta^K$. *The block matrix* $\mathbf{H} \in \mathbb{R}^{dK}$:

$$
\mathbf{H}_{j,j'} = \begin{cases} (\gamma_j - \gamma_j^2)\mathbf{g}_j\mathbf{g}_j^\top \\ -\gamma_j\gamma_{j'}\mathbf{g}_j\mathbf{g}_{j'}^\top & \text{when } j \neq j', \end{cases}
\tag{52}
$$

*is positive semi-definite (PSD). If in addition* $\|\mathbf{g}_j\| \leq B < \infty, \forall j \in [K]$, *then* $\mathbf{H} \preccurlyeq B^2 \cdot I_{dK}$

*Proof.* Let $\mathbf{x} = [\mathbf{x}_1, \ldots, \mathbf{x}_K] \in \mathbb{R}^{dK}$, then

$$
\mathbf{x}^\top \mathbf{H}\mathbf{x} = \sum_{j,j'=1}^K \mathbf{x}_j^\top \mathbf{H}_{j,j'}\mathbf{x}_j
\tag{53}
$$

$$
= \sum_{j=1}^K \left( \mathbf{x}_j^\top \mathbf{H}_{j,j}\mathbf{x}_j + \sum_{j'\neq j} \mathbf{x}_j^\top \mathbf{H}_{j,j'}\mathbf{x}_{j'} \right)
\tag{54}
$$

$$
= \sum_{j=1}^K (\gamma_j - \gamma_j)^2 \cdot (\mathbf{x}_j^\top \mathbf{g}_j)^2 - \sum_{j=1}^K \left( \sum_{j'\neq j} \gamma_j\gamma_{j'} \cdot (\mathbf{x}_j^\top \mathbf{g}_j) \cdot (\mathbf{x}_{j'}^\top \mathbf{g}_{j'}) \right)
\tag{55}
$$

$$
= \sum_{j=1}^K \gamma_j(1 - \gamma_j) \cdot (\mathbf{x}_j^\top \mathbf{g}_j)^2 - \sum_{j=1}^K \left( \gamma_j(\mathbf{x}_j^\top \mathbf{g}_j) \cdot \sum_{j'\neq j} \gamma_{j'} \cdot (\mathbf{x}_{j'}^\top \mathbf{g}_{j'}) \right)
\tag{56}
$$

$$
= \sum_{j=1}^K \gamma_j \left( \sum_{j'\neq j} \gamma_{j'} \right) \cdot (\mathbf{x}_j^\top \mathbf{g}_j)^2 - \sum_{j=1}^K \left( \gamma_j(\mathbf{x}_j^\top \mathbf{g}_j) \cdot \sum_{j'\neq j} \gamma_{j'} \cdot (\mathbf{x}_{j'}^\top \mathbf{g}_{j'}) \right)
\tag{57}
$$

$$
= \sum_{j=1}^K \gamma_j(\mathbf{x}_j^\top \mathbf{g}_j) \cdot \sum_{j'\neq j} \gamma_{j'} \left( \mathbf{x}_j^\top \mathbf{g}_j - \mathbf{x}_{j'}^\top \mathbf{g}_{j'} \right)
\tag{58}
$$

$$
= \sum_{j=1}^K \gamma_j(\mathbf{x}_j^\top \mathbf{g}_j) \cdot \sum_{j'=1}^K \gamma_{j'} \left( \mathbf{x}_j^\top \mathbf{g}_j - \mathbf{x}_{j'}^\top \mathbf{g}_{j'} \right)
\tag{59}
$$

$$
= \sum_{j=1}^K \gamma_j(\mathbf{x}_j^\top \mathbf{g}_j)^2 - \left( \sum_{j=1}^K \gamma_j\mathbf{x}_j^\top \mathbf{g}_j \right)^2
\tag{60}
$$

$$
= \mathbb{E}_{j\sim\boldsymbol{\gamma}}[(\mathbf{x}_j^\top \mathbf{g}_j)^2] - \left( \mathbb{E}_{j\sim\boldsymbol{\gamma}}[\mathbf{x}_j^\top \mathbf{g}_j] \right)^2
\tag{61}
$$

$$
= \mathbb{V}_{j\sim\boldsymbol{\gamma}}[\mathbf{x}_j^\top \mathbf{g}_j] \geq 0,
\tag{62}
$$

where we have repeatedly applied $\sum \gamma_j = 1$ and $\mathbb{E}, \mathbb{V}$ denote expectation and variance, treating $\mathbf{x}_j^\top \mathbf{g}_j$ as a random variable. As a result, $\mathbf{H}$ is PSD.

Suppose in addition $\|\mathbf{g}_j\| \leq B < \infty, \forall j \in [K]$. Using the Cauchy-Schwarz inequality, we have

$$-B \cdot \|\mathbf{x}_j\| \leq -\|\mathbf{x}_j\| \cdot \|\mathbf{g}_j\| \leq \mathbf{x}_j^\top \mathbf{g}_j \leq \|\mathbf{x}_j\| \cdot \|\mathbf{g}_j\| \leq B \cdot \|\mathbf{x}_j\|. \tag{63}$$

Since $\|\mathbf{x}_j\| \leq \|\mathbf{x}\|, \forall j \in [K]$, we have

$$-B \cdot \|\mathbf{x}\| \leq \mathbf{x}_j^\top \mathbf{g}_j \leq B \cdot \|\mathbf{x}\|. \tag{64}$$

Finally, with the Popoviciu's inequality on variances, we have

$$\mathbf{x}^\top \mathbf{H} \mathbf{x} = \mathbb{V}_{j \sim \gamma}[\mathbf{x}_j^\top \mathbf{g}_j] \leq \frac{1}{4}(B \cdot \|\mathbf{x}\| + B \cdot \|\mathbf{x}\|)^2 = B^2 \|\mathbf{x}\|^2, \tag{65}$$

which means $\mathbf{H} \preccurlyeq B^2 I_{dK}$. $\qquad\square$

As shown above, our algorithm can be seen as an instantiation of the Federated Surrogate Optimization (Marfoq et al., 2021, Algo.3). As a result, we have the following convergence derived from Marfoq et al. (2021, Thm.3.2′).

**Theorem 3.1.** *[Convergence] Under Assumptions B.1-B.6, when the clients use SGD with learning rate* $\eta = \frac{a_0}{\sqrt{T}} > 0$, *and the number of rounds* $T \geq a_0^2 \cdot \max\{8L^2, 16L^2\beta^2\}$, *the iterates of our algorithm satisfy*

$$\frac{1}{T} \sum_{t=1}^{T} \mathbb{E}\|\nabla_\Phi f(\Phi^t, \Pi^t)\|_F^2 \leq \mathcal{O}\left(\frac{1}{\sqrt{T}}\right), \tag{7}$$

$$and \quad \frac{1}{T} \sum_{t=1}^{T} \mathbb{E}[\Delta_\Pi f(\Phi^t, \Pi^t)] \leq \mathcal{O}\left(\frac{1}{T^{3/4}}\right), \tag{8}$$

*where the expectation is over the random batch samples and* $\Delta_\Pi f(\Phi^t, \Pi^t) := f(\Phi^t, \Pi^t) - f(\Phi^t, \Pi^{t+1}) \geq 0$.

## C   Additional Experiment Details

**Dataset**   To speed up training, we take 10%, and 15% of the training data from CIFAR-10, and CIFAR-100 respectively. For the Office-Home dataset, we merge images from all domains to get the training dataset, and use the features extracted from the penultimate layer of ResNet-18 pretrained on ImageNet.

**Models and Methods**   For CIFAR-10, we use the CNN2 from Shen et al. (2020) with three 3×3 convolution layers (each with 128 channels followed with 2×2 max pooling and ReLu activation) and one FC layer. For CIFAR-100, we use ResNet-18 as in Marfoq et al. (2021). For Office-Home, the model is an MLP with two hidden layers (1000 and 200 hidden units). The batch size is 50 for CIFAR, and 100 for Office-Home. For FedFomo, we use 5 local epochs in CIFAR-100 to adapt to the noisiness of training and 1 local epoch per communication round for all other experiments.

**Settings**   CIFAR experiments use 8 clients and Office-Home experiments use 10 clients.

**Computational resources and software**   We summarize the computational resources used for the experiments in Table 2 and software versions in Table 3.

Table 2: Summary of computational resource

| Memory | CPU | GPU |
|---|---|---|
| 700GB | Intel(R) Xeon(R) Platinum 8168@2.70GHz | 8 Tesla V100-SXM2 |

Table 3: Software versions

| Operating System | Python | Pytorch | mpi4py |
|---|---|---|---|
| Ubuntu 18.04.5 | 3.9 | 1.9.0 | 3.1.2 |

## D  Additional experimental results

**Client weights with accumulative loss**   Figure 8 provides more complete information compared to Figure 5 from Section 4.4 of the main text. It shows the client weights over training for each of the 8 clients using the accumulative loss (only client 0 and 1 were shown in Figure 5). We find that clients 0, 3, 4, and 5, who share a common data distribution, all end up using client 5's model only, whereas clients 1, 2, 6, and 7 end up using client 6's model only. Hence, although the client weight vectors become one-hot part way through training, clients are not learning in isolation, and are still benefiting from their collaborators.

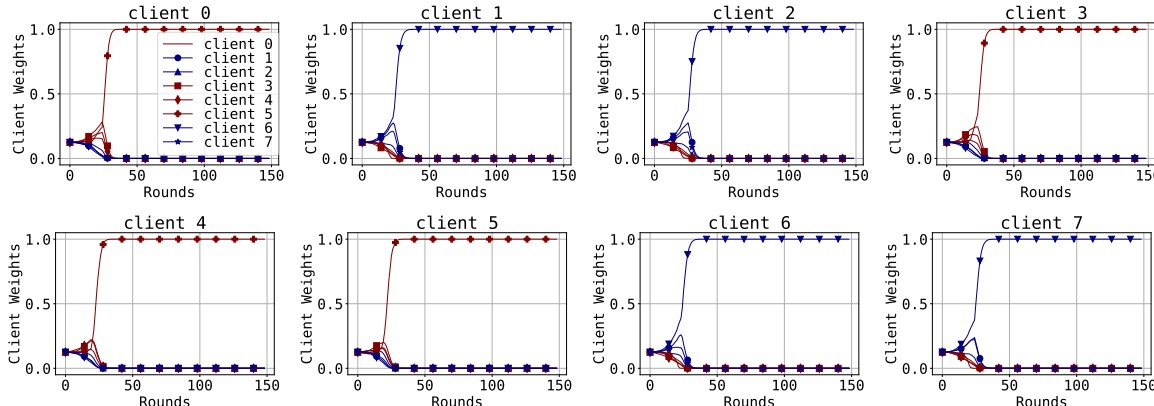

Figure 8: Client weights over time of FedeRiCo with accumulative loss.

**Dirichlet data split**  In Table 4 we compare FedeRiCo with the other baselines with Office-Home dataset using a different data splitting approach. Specifically, we firstly partition the data labels into 4 clusters and then distribute data within the same clusters across different clients using a symmetric Dirichlet distribution with parameter of 0.4, as in FedEM (Marfoq et al., 2021)[6]. As a result, each client contains a slightly different mixture of the 4 distributions. The results are reported over a single run.

| Method | FedAvg | FedAvg+ | Local Training | Clustered FL | FedEM | FedFomo | FedeRiCo |
|---|---|---|---|---|---|---|---|
| **Accuracy** | 69.73±11.02 | 71.20±24.41 | 68.32±19.43 | 69.73±11.02 | 47.15±25.43 | 75.78±6.20 | **83.90±4.11** |

Table 4: Accuracy of different algorithms with Office-Home dataset and Dirichlet distribution.

**Client collaboration**  Here we include more client weight plots, similar to Figure 3 in the main text, of our proposed FedeRiCo on CIFAR100 with four client distributions to show how the level of collaboration evolves over training. As shown in Figure 9, clients from the same distribution have higher client weights and hence more collaboration as expected.

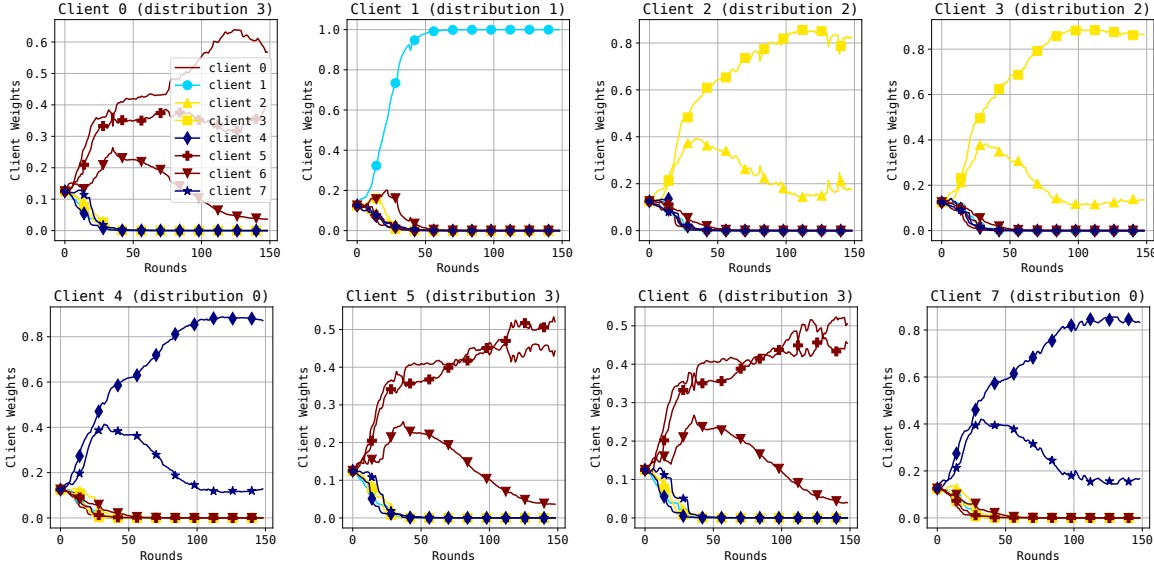

Figure 9: Client weights over time of FedeRiCo with CIFAR100 data and four different client distributions. Clients are color coded by their private data's distribution.

---

[6]We use the implementation from `https://github.com/omarfoq/FedEM`.

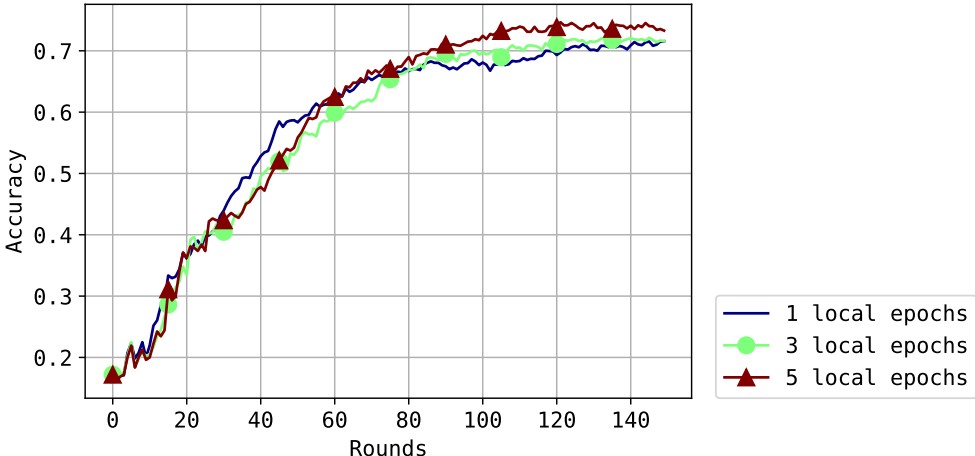

Figure 10: Accuracy vs number of local epochs.

**Number of local epochs** We compared the average training accuracy over clients for different amounts of optimization in each M-step. As shown in Figure 10, when the number of local epochs is increased, and the M-step is better optimized, the test accuracy can converge to a better result. This can be characterized as a trade-off between performance and training time.

