# OpenReview forum: "Find Your Friends: Personalized Federated Learning with the Right Collaborators"
_TMLR — Rejected by TMLR_

### Review · Reviewer_g7wt · 2023-06-12

**Summary Of Contributions:**

The paper proposes a variant to personalized federated learning where local clients compute a weighted average of the gradients of other clients. The weights are determined by the loss of other client's models on its own data.

**Audience:**

Yes

**Broader Impact Concerns:**

I have no concerns about the broader impact.

**Claims And Evidence:**

No

**Requested Changes:**

- in the introduction, the paper argues that "forcing everyone to use the same global model without proper personalization can hurt performance on their own data distribution". This is exemplified using a set of linear models for parts of a non-linear curve: each local model can fit its part well, but there is no good overall linear model for the curve. While this argument makes sense for hypothesis classes of limited capacity, it seems unjustified in the interpolation regime, where for every (practical) dataset there exists at least on hypothesis that reaches negligible empirical error - this encompasses a large part of deep learning. In this setup one could argue that there exists a global model that achieves optimal performance across all local clients. Of course, it might be impractical to find it via federated learning, but I think this argument needs to be strengthened, because in its current form it appears to be wrong.
- In section 3.2 the paper argues that "every client's local distribution $D_i$ can always be represented as a mixture". Without further assumptions, this is wrong. Assume some input $X=\{1,2,3\}$, then already for the marginal over $X$ there is a simple counter example. Assume for $3$ clients local distributions $p_i(x)=1,$ if $x=i$ and $0$ otherwise. Then any $p_i$ cannot be represented as a mixture of the other two. This is a serious problem, since the entire paper hinges on this assumption. Moreover, it appears to me that without such explicit assumptions, the impossibility result in Marfoq, et al. 2021 should apply to this paper as well and no gain over local learning plus a constant should be possible.
- In remark 2 in Section 3, the paper states that Marfoq, et al. 2021 (NeurIPS) differs from this work in that every data point comes from a mixture of distributions. This seems to be a misrepresentation of the work of Marfoq, et al. 2021, which has a similar setting as this work: each client has an individual data distribution $D_t$ which is a mixture of a set of base distributions. The main difference is that Marfoq, et al. 2021 make this assumption explicit, while this paper assumes that each client's distribution is a mixture of the other client's distributions. This work even uses the assumptions from Marfoq, et al. 2021 for its convergence proof. I would argue that a more thorough discussion on the similarities and differences between these two works is necessary. This should include a discussion on the difference in experimental setup, since Marfoq, et al. 2021 reports consistently higher performance on CIFAR10 and CIFAR100 (in both cases higher than the performance achieved by the method proposed in this paper), while the results of local training match with those reported in this work for $4$ distributions.
- An often cited benefit of federated learning over other means of collaboration is its privacy benefit, which is not addressed in this paper.  It would be great to have a clear discussion about the privacy issues of sharing all models with all clients.
- The paper provides several tweaks to the approach to improve communication-efficiency. The paper, however, provides no theoretical or empirical evaluation of the communication complexity. Since communication efficiency is one of the selling points of federated learning, such an evaluation seems necessary.
- The performance comparison in table 1 lacks a centralized baseline. Since a main claim of the paper is that a single, centralized model is inferior to personalized local models, this comparison is essential.

**Strengths And Weaknesses:**

Strength:
- personalized federated learning is an important problem
- the paper proves convergence of the proposed method

Weaknesses:
- the setting of this paper is not perfectly sound (see requested changes)
- the differences to the work of Marfoq, et al. 2021 are not clear
- the paper does not discuss and evaluate privacy aspects and communication complexity of the proposed approach

---

> ### Author Response · Authors · 2023-06-20
> **Reply to reviewer  g7wt [1/3]**
>
> We thank the reviewer for the valuable feedback and the many thoughtful questions. We will address them one by one.
>
> **In the introduction, the paper argues that "forcing everyone to use the same global model without proper personalization can hurt performance on their own data distribution"... it appears to be wrong.**
>
> We believe that the reviewer is not sufficiently considering the role and necessity of personalization in FL. Let us consider the point that “for every (practical) dataset there exists at least one hypothesis that reaches negligible empirical error”. This refers to error on the test set, which in standard FL (FedAvg) is usually assumed to come from the same distribution as every client’s training data. If all clients have data from the same distribution, then effectively increasing the training dataset size through FL’s data sharing may be enough to reach the interpolating regime and train a central model that “achieves optimal performance across all local clients” as the reviewer says. Individual clients may not have sufficient data to reach the interpolating regime on their own, which shows the benefit of FL.
>
> However, our setting is Personalized FL, where the assumption that clients all have data from the same distribution does not hold. This is a much more realistic assumption for many applications and has been a focal point of FL research in the literature. The reviewer quoted a very relevant piece of Section 3.1 in this question, but allow us to reproduce the entire sentence. We write “If [the clients’] **data distributions are vastly different**, forcing them to collaborate is likely to result in worse performance compared to local training without collaboration.” Let us consider how standard FL would perform in this regime using the assumption suggested by the reviewer. A centralized test set would be constructed by sampling from each clients’ distribution and merging, but this joint test set would not be from the same distribution as any individual client’s training data. Even if we can reach the interpolating regime because of FL’s data sharing and find a hypothesis that achieves negligible error on the joint test set, that model will not “achieve optimal performance across all local clients” because their distributions differ. The point of Personalized FL is that no single central model can simultaneously perform optimally on multiple distinct distributions, and hence local personalized models must be used.
>
> In the toy example shown in Figure 1 all clients have input distributions that are essentially separable, so one complex model fitting the sine curve could work for all clients simultaneously. More realistically some clients would have overlapping input distributions but would require different predictions for the same inputs. This is the case discussed in the previous paragraph when no central model could perform well on all clients simultaneously.
>
> **In section 3.2 the paper argues that "every client's local distribution  D_i can always be represented as a mixture"...should be possible**
>
> We would like to clarify the misunderstanding of our assumptions. In our paper, specifically in Section 3.2, we explicitly state that each client's distribution $\mathcal{D}_i$ is a mixture of $ \\{ \mathcal{D}_j \\} _{j=1}^K$, where the index runs over all clients, including client $i$ itself. Therefore, our stated assumption is that every client can be represented as a mixture of **all clients’ distributions, including itself**. This is trivially true, since the one-hot mixture is always feasible, even when client distributions are completely different. However, non-trivial solutions may also exist when some clients are similar to each other, in which case they may use more diverse coefficients that promote collaboration. In extreme settings where client distributions are vastly different from each other there would be no benefit to collaboration, and local learning would be the only feasible solution, represented by the one-hot mixture.
>
> We show empirically that when similar clients exist, collaboration is encouraged by FedeRiCo and better results are obtained compared to the one-hot mixture , as shown in Table 1 (compare Local Training vs. FedeRiCo).

---

> > ### Author Response · Authors · 2023-06-20
> > **Reply to reviewer g7wt [2/3]**
> >
> > **In remark 2 in Section 3, the paper states that Marfoq, et al. 2021 (NeurIPS) differs from this work in that every data point... 4 distributions**
> >
> > Since our approach builds off of FedEM, we took extra care to compare the approaches in Paragraph 4 of Section 2, Remark 2 of Section 3, as well as making several comparisons of their empirical results in Section 4. Both FedEM and FedeRiCo formulate the problem of personalized federated learning as an E-M optimization problem, with a latent variable assigning the data distribution on each client. However, there are subtle but important differences between the assumptions, implementations, and empirical results of these two methods.
> >
> > ### Theoretical and implementation differences
> > - In FedEM, it is assumed that there are $M$ underlying independent distributions, and each data point is generated from one or a combination of these distributions. This implies that the latent assignment is done at the instance level, as demonstrated in equations 8 and 10 in the original FedEM paper (with a superscript (i) to indicate each instance). During the M-step of FedEM, the parameters of each component mode are updated, taking into account the different weights assigned to each data point from the same client for the same component.
> >
> > - In FedeRiCo, it is assumed that all data points belonging to a client come from the same distribution, while the distribution itself is a mixture of the data distributions from all clients. This means that the latent assignment is performed at the client level, as illustrated in equations 3 and 4 in our paper (without any superscript to indicate instance). During the M-step of FedeRiCo, the parameters of each client model are updated, considering the same weights assigned to data points from the same client for the corresponding client model.
> >
> > Consequently, the empirical results show distinct behaviors between FedeRiCo and FedEM. The discrepancy in FedEM stems from the utilization of instance-level component weights during the training phase and client-level component weights during inference. This mismatch can result in a considerable reduction in classification accuracy and a greater variance when making predictions on new test data. Table 1 provides a demonstration of these effects when comparing FedeRiCo and FedEM.
> >
> > ### Experimental setup difference
> > - To create a scenario where the similarity between clients is known, our experiments followed a specific setup. Initially, we divided the entire dataset into multiple distributions, with each client belonging to a single distribution. In this configuration, clients from the same distribution were considered favorable collaborators. As the training progressed, we observed that with FedeRiCo (as depicted in Figure 3), the similar clients effectively learned to collaborate with each other. However, in contrast, FedEM failed to identify the distinct underlying distributions, even when utilizing four component models (as shown in Figure 4). Surprisingly, FedEM assigned the same component model to two different client distributions (distribution 1 and 3) and disregarded component 0 altogether.
> >
> > - In contrast to FedeRiCo, the FedEM experiment utilized a different experimental setting. The dataset was initially partitioned into $k$ clusters, where each cluster represented a distinct label. Subsequently, each client received a proportion of data from each class based on a Dirichlet sampling. Consequently, each client's data became a mixture of $k$ distributions, with $k$ representing the number of classes. Unlike our setting where the favorable collaborators are clear, this setting made it challenging to identify similar clients and observe their collaboration. We also reported empirical results with this data split in Table 4 in the appendix. FedeRiCo consistently outperformed other methods, further showcasing its effectiveness and robustness.

---

> > > ### Author Response · Authors · 2023-06-20
> > > **Reply to reviewer g7wt [3/3]**
> > >
> > > **An often cited benefit of federated learning over other means of collaboration is its privacy benefit, which is not addressed in this paper. It would be great to have a clear discussion about the privacy issues of sharing all models with all clients.**
> > >
> > > Please note that we did discuss privacy implications on page 12.
> > >
> > > While we agree that FL is often cited as providing privacy benefits, we note that it does not provide any guarantee of privacy, and extensive research has shown shared models are still vulnerable to attacks including membership inference and feature extraction. We do not view FL as a method for providing privacy, but as a method for training models on distributed data. Privacy guarantees can be incorporated if required using differential privacy.
> > >
> > > In principle, sharing unprotected models with other clients is no different than sharing models with a central server. In each case a client must either protect their own information (for example by using differential privacy), or the client must trust the receiving party (parties).
> > >
> > > **The paper provides several tweaks to the approach to improve communication-efficiency. The paper, however, provides no theoretical or empirical evaluation of the communication complexity. Since communication efficiency is one of the selling points of federated learning, such an evaluation seems necessary.**
> > >
> > > We omitted a full discussion of communication efficiency because there is essentially no difference between FedeRiCo and standard approaches to FL. To make this more clear, we could add the points below if the reviewer finds them helpful:
> > >
> > > In the centralized setting of FedeRiCo, the communication cost is the same as FedAvg. Each client sends their model to the server, the server computes the relevant gradients, and each client receives one gradient update. In both approaches the central server can become a bottleneck since it may be required to receive and send many updates simultaneously.
> > >
> > > In the decentralized setting of FedeRiCo, we recommended and implemented the version where each client requests one other model per round. On average the communication cost would still be the same as FedAvg, since on average each client sends out one model per communication round.
> > >
> > > There may be a potential communication burden if a single client's model is favored and selected by multiple other clients in a round. However, it is easy to suggest implementation details that can moderate the bottlenecks of any one client receiving too many requests, and which have been explored in the literature to varying extents. If one client receives multiple requests it can seek the assistance of others to relay its model. After another client receives the requested model, that client can begin transferring the model to any others who requested it. Due to the exponential spread as each receiver becomes a sender, broadcasting can be completed in logarithmic time, rather than the linear time implied by the naive approach of one popular client fulfilling many requests on its own.
> > >
> > > **The performance comparison in table 1 lacks a centralized baseline. Since a main claim of the paper is that a single, centralized model is inferior to personalized local models, this comparison is essential.**
> > >
> > > We agree that a centralized baseline is essential, which is why we did already include a centralized baseline (FedAvg) in the first row in Table 1. FedAvg is almost always inferior to local training on the tasks in Table 1, whereas personalized FL approaches, especially FedeRiCo, perform much better.

---

> > > > ### Comment · Reviewer_g7wt · 2023-07-13
> > > > **Response to authors**
> > > >
> > > > **Privacy**
> > > >
> > > > Apologies for my ambiguous formulation. The paper mentions that privacy is an issue in FL and that differential privacy is a common way to address it. It then states that the proposed approach is compatible with differential privacy.
> > > > This statement is not supported. In order to show differential privacy, the sensitivity of the mechanism (in this case the EM-training) has to be bounded and an appropriate noise has to be selected based on this bound to guarantee differential privacy. So while it is surely possible to add noise to models in the same way other FL approaches do, the analysis of the sensitivity is missing.

---

> > > > > ### Author Response · Authors · 2023-07-15
> > > > > **Reply to reviewer g7wt on Privacy**
> > > > >
> > > > > We completely agree with the reviewer that in order to show differential privacy (DP) guarantees, one must bound sensitivity, and add calibrated noise. To be clear, the proposed method FedeRiCo does not by itself satisfy a differential privacy guarantee. When we wrote “most of the existing FL frameworks, including FedeRiCo are compatible with differential privacy” in Section 5, we meant that it is easy to incorporate differentially private training techniques on top of FL methods. For example, DPSGD [B] is the most common way to train ML models with DP guarantees. It takes per-sample gradients, clips them to ensure bounded sensitivity, averages over a batch, and then adds Gaussian noise. DPSGD can be used as a drop in replacement anywhere that SGD is used in federated settings, so we say that existing FL frameworks are compatible with DP. Specifically for FedeRiCo, gradients are computed in the M-step using SGD, and then shared with other clients. By using DPSGD instead, both the client’s gradients and updated model parameters would satisfy DP guarantees which would limit the privacy exposure of their local data when sharing with other clients. We can make these points more explicit in Section 5.
> > > > >
> > > > > [B] Abadi et al. Deep Learning with Differential Privacy, 2016 ACM SIGSAC Conference on Computer and Communication Security

---

> > > ### Comment · Reviewer_g7wt · 2023-07-13
> > > **Reply to authors**
> > >
> > > **Differences to Marfoq, et al. 2021**
> > >
> > > I disagree with the authors about the differences to Marfoq, et al. 2021. In assumption 1 in Marfoq, et al. 2021, it is stated that each client $t$ has a dataset $S_t$ drawn iid. from a distribution $D_t$ that is a mixture of $M$ distributions. Using $M=K$ and having each $D_t$ be one of the $K$ distributions, we arrive at the assumptions of this paper. Therefore, I agree with my fellow reviewers that the scenario of this paper seems to be a special case of Marfoq, et al. 2021.

---

> > > > ### Author Response · Authors · 2023-07-15
> > > > **Reply to reviewer g7wt on Differences to Marfoq, et al. 2021**
> > > >
> > > > Whether or not one scenario is a special case of another does not seem like a critical distinction for discussing the merits of the approaches. Our work and Marfoq et al. 2021 (FedEM) certainly do have similar settings and assumptions. However, the methods have profound differences in terms of implementation and empirical performance.
> > > >
> > > > In FedEM, even though the hidden variable $z$ is stated to be on the client level as in Eq.(2) of that work ($z_t$ with a subscript of $t$ for the client), the actual implementation used a hidden variable on the instance level (see Eq.(8)-(10) of that work, where $z_t^{(i)}$ has an additional superscript $(i)$ for instance). This becomes problematic at inference time since one will not know the hidden variable for a novel data point, and thus cannot perform model averaging for that particular instance. FedeRiCo does not have such a problem because our implementation is consistent with the assumption of client-level hidden variables.
> > > >
> > > > As a further comparison, we have now tested FedEM with 8 components, matching the number of clients. FedEM still shows a lower mean accuracy and higher standard deviation compared to the Local Training and FedeRiCo results (copied from our paper for ease of comparison). Notably, with 8 components, each client would need to store 8 copies of the model and train them each round. Additionally, the aggregation also takes 8N (N=number of clients) updates and sends the updated 8 components to each client, which increases the computation cost and communication cost drastically compared to using a lower number of components. In our original experiments, we chose to use 4 components with FedEM and 3 neighbors for FedeRico in our experiments for a more fair comparison in terms of computational cost. With this setting, in both methods each client trains 4 models every round and sends out the corresponding update so the communication and training costs are comparable.
> > > >
> > > > |          CIFAR-10 # distributions            |             2             |         3        |         4        |
> > > > |----------------------|---------------------------|------------------|------------------|
> > > > | Local Training       | 40.09 $\pm$ 2.84          | 55.27 $\pm$ 3.11 | 69.03 $\pm$ 7.05 |
> > > > | FedEM (8 components) | 55.38 $\pm$ 3.85          | 66.41 $\pm$ 3.09 | 73.33 $\pm$ 7.03 |
> > > > | FedeRiCo             | 56.61 $\pm$ 2.51          | 69.76 $\pm$ 2.25 | 78.22 $\pm$ 4.80 |

---

### Review · Reviewer_3fbB · 2023-06-26

**Summary Of Contributions:**

In this paper, the authors propose a personalized extension of the FedEM method proposed by Marfoq et al. The authors provide convergence guarantee for the proposed method. Empirical results on CIFAR also shows that the proposed method seems to achieve stronger performance.

**Audience:**

Yes

**Claims And Evidence:**

No

**Requested Changes:**

See weaknesses above. In summary, I would like to authors to provide:
- Explanations to me concerns and questions above.
- Reorganize the technical parts so that the notations and assumptions are clear.
- Additional experiments on larger scale datasets and more baselines.

**Strengths And Weaknesses:**

Strengths:
- This work is focusing on personalized federated learning, which is an important area of study.
- The paper organization is clear.

Weaknesses:
1. **Presentation:** I found the presentation unclear in many places which makes understanding the method hard.
- The authors seems to assume the $\mathcal{D}_i$ can be represented as a linear combination of $K$ distributions where $K$ concides with the number of clients. Isn't that a special case of the original Marfoq et al. paper where they assume $M$ distributions?
- In that case, is $\Phi_i$ the personalized model for distribution or the personalized model for client? It seems to be the personalized model for distribution but based on Equation 4, all $K$ models should be broadcast to all the clients, which could introduce gigantic amount of communication in cross-device setting where there are thousands/millions of users.
- The optimization objective is not clearly stated. Could the authors explain what should be minimized / maximized in Equation 2?
2. **Effectiveness:** Given the presentation ambiguity, I don't think I'm convinced by the effectiveness of the current method. It seems that for this personalized optimization problem, the minimizer should converge to every one learns its own local model, i.e. $w_{ij}=1$ iff $i=j$. Please correct me if I'm wrong. Given my understanding above, it is unclear to me why there is a huge difference between local training and the proposed method. If there is any, it might due to that in Algorithm 1, the solver is not finding the exact solution but just doing a gradient update step. However, that would mean the benefits come from ways of optimization instead of the objective itself.
3. **Theory:** Beyond that, could the authors explain how do we benefit from using personalization theoretically compared to learning a global model under mixture of distribution (Marfoq et al.). It seems that the convergence rate is identical. But then it is hard to explain why the empirical difference is large between the two methods. If the authors could provide any explanation that would be great.
4. **Decentralization:** The motivation for decentralization is unclear to me. Why could not we use SMC when we don't have a trusted server? Is decentralization itself a contribution of this work?
5. **Experiments:** Having FedAvg of CIFAR10 to be nearly random guessing seems strange to me. What's more concerning is FedAvg+, which is FedAvg with local finetuning, is much worse compared to local training on CIFAR10, which does not make sense, since in that case FedAvg should just provide local training a different initialization. Could the authors explain why this is the case? Beyond that, it would be interesting to see how the methods could perform in real world cross-device FL cases where you have thousands clients [1]. A bunch of baselines are also missing, including pFedMe [2], Ditto [3], etc.

[1] Caldas, Sebastian, et al. "Leaf: A benchmark for federated settings." arXiv preprint arXiv:1812.01097 (2018).

[2] T Dinh, C., Tran, N., & Nguyen, J. (2020). Personalized federated learning with moreau envelopes. Advances in Neural Information Processing Systems, 33, 21394-21405.

[3] Li, T., Hu, S., Beirami, A., & Smith, V. (2021, July). Ditto: Fair and robust federated learning through personalization. In International Conference on Machine Learning (pp. 6357-6368). PMLR.

---

> ### Author Response · Authors · 2023-07-07
> **Reply to reviewer 3fbB[1/3]**
>
> We thank the reviewer for their valuable comments and would like to address their concerns as follow.
>
> **The authors seems to assume the D_i  can be represented as a linear combination of K distributions where K concides with the number of clients. Isn't that a special case of the original Marfoq et al. paper where they assume M distributions?**
>
> In our setting, described in Sec. 3.1, the $K$ distributions $\mathcal{D}_i$ are potentially distinct, and each is associated with one of the $K$ clients. In Sec. 3.2 we note that each local distribution $\mathcal{D}_i$ can be represented as a mixture of all K distributions. Of course, the trivial case of a one-hot mixture is always possible, but more interesting mixtures may also be possible when some clients are similar.
>
> Marfoq et al. additionally assume that there are $M$ underlying, independent distributions $\tilde{D}_m$ (see beginning of Sec. 2 in their paper), and furthermore that the local distributions can be expressed as mixtures of the underlying distributions (see Assumption 1 in their paper).
>
> Our setting is not a special case of Marfoq et al., it is a generalization in that we do not require Assumption 1 to hold. Note that setting $M=K$ in Assumption 1 is still a non-trivial assumption, whereas our setting is self-evidently true in all reasonable personalized federated learning problems.
>
> **In that case, is Φ_i  the personalized model for distribution or the personalized model for client? It seems ... which could introduce gigantic amount of communication in cross-device setting where there are thousands/millions of users.**
>
> $\Phi$ is used to represent the vector of individual models $\phi_i$ associated with each client. We agree that broadcasting all models to all clients would incur huge communication costs, which is why in Sec. 3.3 we proposed a more communication-efficient protocol, where every client samples models from a few neighbors. We empirically found strong performance using a default value of 3 as the number of neighbours sampled in each round, and this hyperparameter was studied in Figure 6 (middle).
>
> **The optimization objective is not clearly stated. Could the authors explain what should be minimized / maximized in Equation 2?**
>
> At the beginning of Sec. 3.2 we state that our initial objective is to maximize the log likelihood defined in Equation 1 over the joint parameters $\Theta$. Then following the derivation in Appendix A, we get the evidence lower bound (ELBO) in Equation 2 where q is an alternative distribution. Finally, we maximize this lower bound using the EM algorithm.  This process is standard in many EM-like algorithms.
>
> **Effectiveness: Given the presentation ambiguity... that would mean the benefits come from ways of optimization instead of the objective itself.**
>
> We would like to clarify that the minimizers would not always converge to local training with one-hot weightings for all clients. When some clients are similar to others, collaborating with diverse weightings could achieve a better objective value than one-hot weightings (local training) as defined in Equation 2. As a simple example, imagine that a subset of clients actually have data drawn from the same local distribution, i.e $\mathcal{D_{i\in S}}=\mathcal{D}^*$ for some subset of clients $S\in [K]$, but each client has limited training data. It would be advantageous for these clients to merge their datasets, and they would surely achieve lower losses in that case. However, the setting of FL assumes that merging datasets is not feasible. In FedeRiCo, the set of clients can effectively increase the sample size and diversity of their local training datasets through collaboration. This will still improve their local test performance over training alone with limited data. Through proper initialization (we use uniform weights as initialization), and randomized exploration, our algorithm is incentivized to find useful collaborators. On the contrary, when clients are significantly different from all others, the client weights would become one-hot and this reduces to local training. However, it is desirable to have this as an option for cases where client datasets truly are different and collaboration would be harmful. As a point of contrast, some personalized FL methods, such as clustered FL, still force clients to collaborate even when it is detrimental due to severe dataset differences. Finally, as a concrete example, Fig. 3 shows the weights on CIFAR100 where only one client (with a unique distribution) tends to learn by itself while all other clients are collaborating with others to varying degrees, resulting in diverse weights $w_{ij}$. To summarize, the objective of our algorithm encourages collaboration when it is beneficial.

---

> > ### Author Response · Authors · 2023-07-07
> > **Reply to reviewer 3fbB[2/3]**
> >
> > **Theory: Beyond that... If the authors could provide any explanation that would be great.**
> >
> > Since our approach builds off of FedEM, we took extra care to compare the approaches in Paragraph 4 of Section 2, Remark 2 of Section 3, as well as making several comparisons of their empirical results in Section 4. Both FedEM and FedeRiCo formulate the problem of personalized federated learning as an EM optimization problem, with a latent variable assigning the data distribution on each client. However, there are subtle but important differences between the assumptions, implementations, and empirical results of these two methods.
> >
> > In FedEM, it is assumed that there are $M$ underlying independent distributions, and each data point is generated from one or a combination of these distributions. This implies that the latent assignment is done at the instance level, as demonstrated in equations 8 and 10 in the original FedEM paper (with a superscript (i) to indicate each instance). During the M-step of FedEM, the parameters of each component mode are updated, taking into account the different weights assigned to each data point from the same client for the same component.
> >
> > In FedeRiCo, it is assumed that all data points belonging to a client come from the same distribution, while the distribution itself is a mixture of the data distributions from all clients. This means that the latent assignment is performed at the client level, as illustrated in equations 3 and 4 in our paper (without any superscript to indicate instance). During the M-step of FedeRiCo, the parameters of each client model are updated, considering the same weights assigned to data points from the same client for the corresponding client model.
> >
> > Consequently, the empirical results show distinct behaviors between FedeRiCo and FedEM. The discrepancy in FedEM stems from the utilization of instance-level component weights during the training phase and client-level component weights during inference. This mismatch can result in a considerable reduction in classification accuracy and a greater variance when making predictions on new test data. Table 1 provides a demonstration of these effects when comparing FedeRiCo and FedEM.
> >
> > **Decentralization: The motivation for decentralization is unclear to me. Why could not we use SMC when we don't have a trusted server? Is decentralization itself a contribution of this work?**
> >
> > We included a paragraph in section 5 discussing the decentralized FL architecture. One of our contributions is that we developed an efficient and effective decentralized scheme tailored to our algorithm. This is essential since naive implementation in the centralized setting can be computationally expensive as the reviewer also mentioned previously.
> > On the other hand, Secure Multiparty Computation can be less computationally efficient than FL systems because of the large cryptographic overhead required, and may not scale well to the same cross-device settings where FL can be used. SMC certainly is an interesting privacy enhancing technology, but fills a different niche than decentralized FL. Since we were already comparing several variations of FL (centralized, decentralized, personalized) we kept the discussion focussed on these topics rather than branching into SMC as well.

---

> > > ### Author Response · Authors · 2023-07-07
> > > **Reply to reviewer 3fbB[3/3]**
> > >
> > > **Experiments: Having FedAvg of CIFAR10 to be ... A bunch of baselines are also missing.**
> > >
> > > Our experiments followed a specific setup where the entire dataset is divided into multiple distributions, with each client belonging to a single distribution. The datasets and non-IID splits are described at the start of Sec. 4.1. In this configuration, methods that enforce collaboration like FedAvg and clustered FL can hurt the performance, as the clients are drastically different from each other. And with FedAvg+, while the unsuccessful centralized training may act as a different initialization for local training, these “initialization” weights are not random, and may reasonably result in worse performance. Additionally, the fine-tuning for FedAvg+ was only one step, which was a lot less than training locally.
> > >
> > > While we agree with the reviewer that large-scale experiments can be interesting, our computing resources are limited. Our code used for all our experimental results (provided in the supplementary material) was developed with mpi4py such that all clients are working *in parallel*, with live (blocking) communication between clients. This structure is fairly distinct from research repositories developed for experimenting on millions of clients, where clients are trained *serially* with no live communication (client information is written to and read from a shared hard drive). We want to emphasize that FedeRiCo can also work with more clients as in a cross-device FL setting: the limiting factor is our computing resources rather than algorithmic design. As for extra baselines, it is noteworthy that both FedEM and FedFOMO have shown better performance than pFedMe, and we compared FedeRiCo to both of them.

---

### Review · Reviewer_pbCX · 2023-06-28

**Summary Of Contributions:**

In order to tackle data heterogeneity, this paper presents a methodology to perform personalised federated learning (FL), by allowing each client $i \in [K]$ to learn a personalised model $\phi_i$.
To model data heterogeneity, the authors assume that each client's local distribution $D_i$ can be expressed as a mixture of other clients' distributions, that is $D_i = \sum_{j=1}^K \pi_{ij} D_j$, where $\pi_{ij}$ stands for a local weight vector.
Similar to statistical inference methods in mixture models (e.g. Gaussian ones), the authors introduce a set of latent variables representing the distribution from which the data has been generated; and rely on expectation maximisation to infer both model parameters and latent variables.
Convergence guarantees are provided and the benefits of the proposed approach are illustrated on a set of experiments involving 3 data sets.

Compared to prior and related works, the novel contributions brought by this paper are :

* The combination of local weights $\pi_{ij} at the client level with the mixture presentation at the client level. Closest works are Mansour et al., 2020 and Marfoq et al., 2021.
* Empirical assessment of the benefits of the proposed methodology referred to as FedeRiCo.

**Audience:**

Yes

**Broader Impact Concerns:**

N/A.

**Claims And Evidence:**

No

**Requested Changes:**

See my comments in the Weaknesses section above.

**Strengths And Weaknesses:**

Strengths:

* The methodology proposed by the authors is clear.
* The authors pointed out several issues if the naive version of FedeRiCo is used and proposed workarounds to cope with them, such as (i) communication bottleneck to update both the clients' weights (E-step) and the local model parameters and (ii) convergence issues associated to the clients' sampling scheme which has been addressed using a smoothing mechanism.
* The numerical assessment is thorough.

Despite these strengths, I have some major concerns regarding the paper in its current state, see below.

Weaknesses:

* Albeit not the main criteria to accept this submission, the novelty of the proposed methodology is rather incremental with respect to prior works, especially Marfoq et al., 2021.
* The appendix lack rigor and clarity, especially the proof of Theorem 3.1. First, I understand that the main assumptions and the proof techniques B1-6 come from Marfoq et al., 2021 but I would expect at the beginning of Section B a brief recap of the proof organisation. For example, I do not understand why you should verify the three conditions of partial first-order surrogates. Second, while it is indeed mandatory to provide high-level theoretical results in the main paper, I would like to have more rigorous results in the Appendix. Notably, (i) replace "after a sufficient number of rounds $T$" with a clear lower bound on $T$; (ii) replace $O(.)$ in (7) and (8) by exact upper bounds.
* Regarding experimental results, I would like to have figures comparing the different approaches w.r.t. the communication overhead (e.g. size of the vectors to be transmitted from one party to another) and to add HypCluster (Mansour et al., 2020) to the set of competitors as it is a specific instance of the proposed methodology.

---

> ### Author Response · Authors · 2023-07-13
> **Reply to reviewer pbCX [1/2]**
>
> We thank the reviewer for their valuable feedback, and address their questions and concerns as follows.
>
> **Albeit not the main criteria to accept this submission, the novelty of the proposed methodology is rather incremental with respect to prior works, especially Marfoq et al., 2021.**
>
> Since our approach builds off of FedEM, we took extra care to compare the approaches in Paragraph 4 of Section 2, Remark 2 of Section 3, as well as making several comparisons of their empirical results in Section 4. Both FedEM and FedeRiCo formulate the problem of personalized federated learning as an E-M optimization problem, with a latent variable assigning the data distribution on each client. However, there are subtle but important differences between the assumptions, implementations, and empirical results of these two methods.
>
> ## Theoretical and implementation differences
>
> In FedEM, it is assumed that there are $M$ underlying independent distributions, and each data point is generated from one or a combination of these distributions. This implies that the latent assignment is done at the instance level, as demonstrated in equations 8 and 10 in the original FedEM paper (with a superscript (i) to indicate each instance). During the M-step of FedEM, the parameters of each component mode are updated, taking into account the different weights assigned to each data point from the same client for the same component.
>
> In FedeRiCo, it is assumed that all data points belonging to a client come from the same distribution, while the distribution itself is a mixture of the data distributions from all clients. This means that the latent assignment is performed at the client level, as illustrated in equations 3 and 4 in our paper (without any superscript to indicate instance). During the M-step of FedeRiCo, the parameters of each client model are updated, considering the same weights assigned to data points from the same client for the corresponding client model.
>
> Consequently, the empirical results show distinct behaviors between FedeRiCo and FedEM. The discrepancy in FedEM stems from the utilization of instance-level component weights during the training phase and client-level component weights during inference. This mismatch can result in a considerable reduction in classification accuracy and a greater variance when making predictions on new test data. Table 1 provides a demonstration of these effects when comparing FedeRiCo and FedEM.
>
> ## Experimental setup difference
>
> To create a scenario where the similarity between clients is known, our experiments followed a specific setup. Initially, we divided the entire dataset into multiple distributions, with each client belonging to a single distribution. In this configuration, clients from the same distribution were considered favorable collaborators. We note that this setting fits Assumption 1 in Marfoq et al 2021., so one might expect FedEM to be a suitable approach.
>
> As training progressed, we observed that with FedeRiCo (as depicted in Figure 3), similar clients effectively learned to collaborate with each other. However, in contrast, FedEM failed to identify the distinct underlying distributions, even when utilizing four component models (as shown in Figure 4). Surprisingly, FedEM assigned the same component model to two different client distributions (distribution 1 and 3) and disregarded component 0 altogether.
>
> In contrast to the setting above, the experiments in Marfoq et al. 2021 used a dataset that was initially partitioned into $k$ clusters, where each cluster represented a distinct label. Subsequently, each client received a proportion of data from each class based on a Dirichlet sampling. Consequently, each client's data became a mixture of $k$ distributions, with $k$ representing the number of classes. Unlike our setting where the favorable collaborators are clear, this setting makes it challenging to identify similar clients and observe their collaboration. For complete comparison, we also reported empirical results with this data split in Table 4 in Appendix D. FedeRiCo consistently outperformed other methods, further showcasing its effectiveness and robustness.

---

> > ### Author Response · Authors · 2023-07-13
> > **Reply to reviewer pbCX [2/2]**
> >
> > **The appendix lack ... upper bounds.**
> >
> > Thank you for the feedback. We have modified Appendix B to provide a high-level description of the proof at the beginning, and rewritten Theorem 3.1 using more precise language.
> >
> > To address your questions, verifying that the properties of partial first-order surrogates hold is necessary because we want to show that our algorithm can be seen as an instantiation of Federated Surrogate Optimization (Algo.3 of Marfoq et al., 2021) where each client locally optimizes the corresponding surrogate (L4-6 of Algo.3 of Marfoq et al., 2021). Then we can apply the meta theorem (Theorem 3.2’ of Marfoq et al., 2021) to prove our convergence. For our Theorem 3.1, T needs to be larger than $a_0^2 \max\{8 L^2, 16L^2\beta^2\}$ (as per Theorem 3.2’ of Marfoq et al., 2021). Note that we use simplified bounds here because they are derived from the original meta theorem, which also uses simplified bounds even in their appendix.
> >
> > **Regarding the experimental results ...  a specific instance of the proposed methodology.**
> >
> > The following table summarizes the communication overhead for all methods in our experiment, where M is the number of parameters of the model and B is the number of components (FedEM), best-performing models (FedFOMO), or neighbors (FedeRiCo).
> > | Method            | FedAvg | FedAvg+ | Local | Clustered FL | FedEM | FedFOMO | FedeRiCo |
> > |-------------------|--------|---------|-------|--------------|-------|---------|----------|
> > | Comm/Round/Client | M      | M       | None  | M            | BM    | BM      | BM       |
> >
> > We thank the reviewer for suggesting including HypCluster as an additional baseline. Although we were unable to implement and run HypCluster due to time and resource constraints, we are committed to providing the results once they become available.

---

> > > ### Author Response · Authors · 2023-08-07
> > > **Hypcluster results**
> > >
> > > Below are the experimental results of HypCluster on the CIFAR datasets. Our experiments show that HypCluster can achieve reasonable performance. However, it is very sensitive to initializations. The poor performance in the 4-distribution setting of CIFAR100 is mainly because it can have empty clusters (clusters with no client assigned to it) due to hard assignment. In comparison, our method uses soft assignment + stochastic sampling, making it more flexible than HypCluster (similar to how EM is more flexible than Lloyd‘s k-means algorithm in specific spherical GMM settings). We will be happy to discuss further if the reviewer has any additional questions/comments.
> > >
> > >
> > > |          CIFAR-10 # distributions            |             2             |         3        |         4        |
> > > |----------------------|---------------------------|------------------|------------------|
> > > | Local Training       | 40.09 $\pm$ 2.84          | 55.27 $\pm$ 3.11 | 69.03 $\pm$ 7.05 |
> > > | Hypcluster(4 clusters)| 34.52 $\pm$12.14 | 56.19 $\pm$ 6.13| 65.22 $\pm$ 7.42  |
> > > | FedeRiCo             | 56.61 $\pm$ 2.51          | 69.76 $\pm$ 2.25 | 78.22 $\pm$ 4.80 |
> > >
> > >
> > > |          CIFAR-100 # distributions            |             2             |         3        |         4        |
> > > |----------------------|---------------------------|------------------|------------------|
> > > | Local Training       | 16.60 $\pm$ 0.64          | 25.99 $\pm$ 2.38 |31.05 $\pm$ 1.68 |
> > > | Hypcluster(4 clusters)| 34.60 $\pm$ 1.63 | 35.57 $\pm$ 2.74| 22.98 $\pm$ 2.31  |
> > > | FedeRiCo             | 30.95 $\pm$ 1.62          | 39.19 $\pm$ 1.64 | 41.41 $\pm$ 1.07 |

---

### Decision · Action_Editors · 2023-10-04

**Recommendation:** Reject

**Comment:**

I want to apologize to the authors for the significant delay in taking the final decision. This was notably due to one of the reviewers becoming unresponsive (despite many reminders) and never submitting his/her recommendation.

Nevertheless, there was a consensus among the other two reviewers to reject the paper for the reasons explained in the "Claims And Evidence" part above. As Action Editor, after having reviewed the entire discussion and the reviewers' recommendations, I agree with their arguments.

**Audience:**

The topic of (personalized) federated learning is of broad interest to TMLR's audience.

**Claims And Evidence:**

This paper proposes an approach for personalized federated learning which is closely related to the FedEM approach of Marfoq et al. (2021). The authors claim to propose a weaker and more suitable assumption on the client's distributions, and that the resulting objective yields more accurate models. However, several reviewers noted that this assumption in fact appears to be a particular case of the assumption made in Marfoq et al. (2021), namely to take the number of mixture distributions to be equal to the number of clients (this seems to be implicitly recognized by the authors in their last response to reviewer g7wt). In light of this observation (which contradicts the claims made in the paper), the proposed setting gives rise to a somewhat degenerate personalized FL scenario: clients can have arbitrary local distributions (although it is known that some assumptions are needed for personalized FL to be guaranteed to work), and the proposed objective can be trivially minimized by assigning a different model to each client (i.e., set the mixture weight matrix to be a permutation matrix). In their response, the authors argued that other minima can exist (which is true) and that their approach can find them, but their arguments about limited data hold from the generalization point of view, not from the optimization point of view (the algorithm optimizes the objective on the training set). The good empirical performance of the approach thus seems to be a side effect of the proposed solver, but it is not clear why it happens (one reviewer notes that the convergence guarantees of the proposed algorithm are the same as for FedEM). In any case, defining an objective function which has a trivial and undesirable optimum does not seem very principled.

Other minor concerns also remain around the missing comparison of popular personalized FL baselines such as pFedMe and Ditto.